# MVISTA-4D: View-Consistent 4D World Model with Test-Time Action Inference for Robotic Manipulation

**Jiaxu Wang** [* 1]   **Yicheng Jiang** [* 2]   **Tianlun He** [2]   **Jingkai Sun** [3]   **Qiang Zhang** [4]   **Jiahang Cao** [3]   **Zesen Gan** [2]
**Mingyuan Sun** [5]   **Qiming Shao** [† 2]   **Xiangyu Yue** [† 1]

## Abstract

World-model-based imagine-then-act becomes a promising paradigm for robotic manipulation, yet existing approaches typically support either purely image-based forecasting or reasoning over partial 3D geometry, limiting their ability to predict complete 4D scene dynamics. To solve this, this work explores a novel embodied 4D world model that enables geometrically consistent, arbitrary-view RGBD generation: given only a single-view RGBD observation as input, the model "imagines" the remaining viewpoints, which can then be back-projected and fused to assemble a more complete 3D structure across time. To efficiently learn the multi-view, cross-modality generation, we explicitly design cross-view and cross-modality feature fusion that jointly encourage consistency between RGB and depth and enforce geometric alignment across views. Beyond prediction, converting generated futures into actions is often handled by inverse dynamics, which is ill-posed because multiple actions can explain the same transition. We address this with a test-time action optimization strategy that backpropagates through the generative model to infer a trajectory-level latent best matching the predicted future, and a residual inverse dynamics model that turns this trajectory prior into accurate executable actions. Extensive experiments on the three datasets and platforms demonstrate strong performance on both 4D scene generation and downstream manipulation, and ablations provide practical insights into the key de-

---

[*]Equal contribution   [1]MMLab, The Chinese University of Hong Kong, Hong Kong SAR [2]The Hong Kong University of Science and Technology, Hong Kong SAR [3]The University of Hong Kong [4]X-Humanoid Robots [5]Tsinghua University. Correspondence to: Qiming Shao <eeqshao@ust.hk>, Xiangyu Yue <xyyue@cuhk.edu.hk>.

*Proceedings of the 43rd International Conference on Machine Learning*, Seoul, South Korea. PMLR 306, 2026. Copyright 2026 by the author(s).

sign choices. Project page is available at https://mercerai.github.io/MVISTA-4D/.

## 1. Introduction

World models are increasingly important for robotic manipulation because they enable predictive decision-making beyond reactive control (Liang et al., 2024; Feng et al., 2025). Rather than directly mapping observations to actions, a world model forecasts future observations under instructions, and the robot chooses actions that best realize desired imagined futures. This "imagine-then-act" paradigm has shown strong gains in planning and long-horizon control.

Most existing world models operate in image space (Zhang et al., 2024; Yan et al., 2021), benefiting from strong video-generation priors. However, manipulation unfolds in the 3D physical world, where geometry and spatial relations govern contact and precise interaction. As a result, pixel-level predictions can appear plausible while violating geometric constraints, creating a gap between imagined futures and executable actions. This has motivated recent efforts toward 4D world models (Wang et al., 2024b; Sun et al., 2024). One line extends video generation to RGB-D, forecasting depth together with appearance (Zhen et al., 2025), but many methods remain single-view, leading to incomplete geometry and brittleness under occlusion, while monocular depth can suffer from scale and temporal drift. Another line models dynamics directly in 3D using point clouds or particle-like representations (Huang et al., 2025b), which are more explicitly geometric but often sparse and provide weaker appearance and semantic cues, limiting fidelity for fine-grained manipulation. Together, these limitations motivate a world model that retains strong visual priors while producing geometry-consistent, multi-view 4D predictions.

A second, equally critical question is: *once we can imagine, how do we convert imagination into actions?* Three paradigms are common. (1) Joint prediction of future observations and actions (Li et al., 2025; Zhou & Negrut, 2025) tightly couples control with generation, but errors in early predicted actions can shift the rollout distribution and compound under off-distribution interactions. (2) Geometry-first

pipelines (Liu et al., 2025; Mao et al., 2025) recover poses from imagined geometry using a pose estimator (Wen et al., 2024) and then derive actions, but they inherit the fragility of pose estimation under occlusion, contact, and geometry artifacts. (3) Inverse dynamics models map consecutive observations to actions (Du et al., 2023; Zhen et al., 2025), yet this formulation is inherently ill-posed: many actions can explain similar perceptual changes, especially under partial observability and contact. Crucially, these paradigms often treat actions as per-step signals and overlook trajectory-level structure, whereas manipulation trajectories lie in a low-dimensional space with strong temporal correlations shaped by kinematics, smoothness, and task constraints.

We address these challenges with a trajectory-conditioned, geometry-consistent multi-view 4D generative world model that synthesizes dynamic manipulation scenes in RGBD from queried viewpoints. We introduce explicit cross-modal and cross-view fusion to improve appearance–geometry consistency and enforce geometry-aligned information flow across views. For action inference, we represent the entire action trajectory with a compact latent code to capture trajectory-level structure and temporal coherence. Given a predicted 4D future, we infer actions by optimizing this trajectory latent through backpropagation, and further refine the resulting sequence with a residual inverse dynamics model for reliable execution. Overall, the contributions can be summarized as follows:

- We propose an embodied multi-view 4D generative world model that produces cross-view and cross-modal consistent future predictions, improving spatiotemporal coherence and providing stronger geometric cues for action inference under occlusion for embodied manipulation.
- We introduce trajectory-level action conditioning by representing an entire action sequence with a compact low-dimensional latent code, capturing task-relevant temporal structure and enabling action inference from generated 4D futures via backpropagation.
- We propose a prior-based residual inverse dynamics model for action refinement, which treats the inferred trajectory as initialization and learns only small correction terms, alleviating the ill-posedness of classic IDM and improving robustness for execution.
- We collect a real-robot 4D multiview dataset with 14 tasks. Extensive experiments demonstrate that our method generates high-fidelity 4D scenes and consistently outperforms strong baselines on downstream manipulation tasks, validating both the effectiveness of the approach and the necessity of our key design choices.

## 2. Related Work

**Video Generative Models**. Building on the success of image generation, many works have extended *generative mod-*

*els* to video by introducing temporal modeling (Yan et al., 2021; Ho et al., 2022) and operating in a latent space for efficient synthesis (Zheng et al., 2024; Yang et al., 2024b). More recently, camera-controlled video generation injects camera parameters to steer viewpoint changes over time. For example, (Bai et al., 2025; Zhang et al., 2025) translate a reference video into novel views with different camera motions. Despite impressive visual quality, these models typically target RGB videos and do not explicitly enforce geometry-consistent multi-view outputs required by contact-rich manipulation.

**World Models for Robot Manipulation**. Motivated by advances in video generation, many works adopt generative world models for robotics, either as auxiliary components for policy learning (Lu et al., 2024; Chai et al., 2025; Yu et al., 2025; Wang et al., 2024a) or as planning modules that imagine task rollouts and then infer actions (Tian et al., 2024; Liang et al., 2025; Bharadhwaj et al., 2024; Huang et al., 2025a). Representative approaches infer actions via inverse dynamics (Du et al., 2023; Zhen et al., 2025), regress actions from intermediate world-model features (Bai et al., 2025), or introduce motion-relevant masks to facilitate action inference (Feng et al., 2025). However, most methods rely on 2D generation, while manipulation fundamentally depends on 3D geometry, creating a gap between imagined futures and executable actions. Recent 3D-aware attempts like TesserAct (Zhen et al., 2025) generate RGB-DN sequences and use a point-based inverse dynamics model, while 4DGen (Liu et al., 2025) estimates pose changes (e.g., via FoundationPose) from generated outputs and maps them to robot commands. Still, many prior methods require training additional action modules, and inverse dynamics remains ill-posed under partial observability and contact. In contrast, we embed the action trajectory directly into generation via a compact trajectory code, recover actions by optimizing this trajectory condition through backpropagation, and further refine them with a residual inverse dynamics model for reliable execution.

**4D Video Generation**. 4D video generation has gained increasing attention in recent years. Early works distill 4D geometry by combining video generation with explicit 3D representations (e.g., NeRF (Mildenhall et al., 2021) or 3DGS (Kerbl et al., 2023)) via SDS-style objectives (Ren et al., 2023; Bahmani et al., 2024; Yin et al., 2023), but often require slow per-scene optimization and can be unstable. Another line decouples temporal consistency (video models) from spatial consistency (novel view synthesis) (Sun et al., 2024; Yang et al., 2024a; Wang et al., 2024b), which may lead to objective mismatch and imperfect spatiotemporal alignment. More recently, methods attempt to generate 4D content more directly. For instance, (Zhen et al., 2025) reconstructs 4D scenes from RGB-DN videos but still needs post-optimization and is limited to single-view outputs, re-

sulting in incomplete geometry. (Liu et al., 2025) produces consistent two-view pointmaps under the DUSt3R paradigm, but it does not generalize well to more views and can be sensitive to the chosen source view.

## 3. Preliminaries

**Imagine-then-act Paradigm**. Unlike the classic *observation to action* paradigm, *imagine then act* first uses a world model to predict instruction-related future observations and then derives actions. At time $t_0$, the agent receives an instruction $l$ and an initial observation $o_0$ with one or multiple RGB or RGB-D views, and predicts

$$\hat{o}_{1:T} = p_\theta(o_{1:T} \mid o_0, l). \tag{1}$$

"Many recent world models adopt video-generation backbones. However, manipulation requires accurate and temporally consistent 3D relations, while $o_0$ is often partial due to limited viewpoints and occlusions. Consequently, $\hat{o}_{1:T}$ can miss task-critical states, undermining action reasoning.

The *act stage* produces actions $a_{0:T-1}$ consistent with $\hat{o}_{1:T}$. Prior methods either add an action head sharing the latent space with $\hat{o}_{1:T}$, increasing complexity, or learn inverse dynamics $a_t = h_\psi(\hat{o}_t, \hat{o}_{t+1})$, which is ill-posed. Both degrade under partial $\hat{o}_{1:T}$. We address this by predicting complementary views with cross-view consistent appearance and geometry, and by injecting a low-dimensional trajectory style code that supports test-time backpropagation to recover actions consistent with the imagined future.

**Latent Video Diffusion Models**. Our study is built on a latent video diffusion framework (Wan et al., 2025) consisting of a 3D VAE and a Transformer-based diffusion model with Flow Matching (Lipman et al., 2023) schemes. The forward process linearly interpolates between data and Gaussian noise, $z_t = (1 - t)z_0 + t\epsilon, \epsilon \sim \mathcal{N}(0, I), t \in [0, 1]$. Denoising is formulated as a probability flow ODE, $\frac{dz_t}{dt} = v_\Theta(z_t, t)$, where $v_\Theta$ is parameterized by $\Theta$. Training regresses the predicted velocity to the target field induced by the linear path,

$$\mathcal{L}_{\text{diff}} = \mathbb{E}_{t, z_0, \epsilon} \|v_\Theta(z_t, t) - (\epsilon - z_0)\|_2^2, \tag{2}$$

with $z_t = (1 - t)z_0 + t\epsilon$. At inference, we solve the ODE with Euler steps. Starting from $z^{(0)} \sim \mathcal{N}(0, I)$, we iterate $z^{(k+1)} = z^{(k)} + v_\Theta(z^{(k)}, t_k)\Delta t$ for $k = 0, \ldots, K-1$, with $t_k = k\Delta t$ and $\Delta t = 1/K$, and decode $z^{(K)}$ with the VAE.

## 4. Methodology

**Problem Statement**. We study *multi-view, spatiotemporally consistent RGB-D generation* for embodied manipulation. Given a single observation $\mathbf{o}_0 = (\mathbf{I}_0, \mathbf{D}_0)$ from a reference view with known extrinsics $\mathbf{T}_0 \in SE(3)$, where

$\mathbf{I}_0 \in \mathbb{R}^{H \times W \times 3}$ is the RGB image and $\mathbf{D}_0 \in \mathbb{R}^{H \times W}$ is the depth map, we also provide a set of target camera extrinsics $\{\mathbf{T}_i\}_{i=1}^N$ and a text instruction $l$. Our goal is to learn a conditional generator $G$ that produces an instruction-following future RGB-D sequence in the reference view, together with synchronized RGB-D sequences for all target views, such that all views depict the same underlying scene dynamics and remain geometrically consistent. The generated multiview RGB-D frames can be back-projected and fused using $\{\mathbf{K}, \mathbf{T}_i\}$ to obtain point-cloud sequences of the dynamic scene. We further infer actions from generated futures via a trajectory latent and residual IDM.

**Input Format Strategy**. We adopt a structured tokenization that reduces token distance for key dependencies after flattening, which helps Transformers model correlations more effectively (Vig & Belinkov, 2019). We directly follow the fusion analysis in 4DNeX (Chen et al., 2025b) and use axis-wise concatenation to control token neighborhoods: width-wise concatenation makes cross-map tokens within the same row adjacent, while height-wise concatenation makes rows from different maps consecutive in token order. Accordingly, within each view, we use width-wise fusion to place RGB–D tokens at the same spatial location next to each other, encouraging cross-modality consistency. Across views, pixel-level alignment is ill-posed due to parallax and occlusion, so we adopt height-wise fusion to promote structure-level cross-view coherence. This layout mainly facilitates cross-view information exchange; explicit geometric correspondence is handled by our geometry-aware cross-view module.

Concretely, for each view $v$ we encode $\mathbf{I}^{(v)}$ and $\mathbf{D}^{(v)}$ with a VAE tokenizer to obtain latent maps $\mathbf{z}^{I,(v)}$ and $\mathbf{z}^{D,(v)}$. We fuse modalities within each view via width-wise concatenation, fuse views via height-wise concatenation, and then flatten the resulting latent tensor into a one-dimensional token sequence for the Transformer generator. This design naturally supports a variable number of input views by simply concatenating along the height dimension.

### 4.1. Feature Integration Across Modalities

**Learnable Modality Token**. Let $\mathbf{x}_i^{\text{app}}, \mathbf{x}_i^{\text{geo}} \in \mathbb{R}^{N \times 1}$ denote the token features at spatial index $i$ for appearance and geometry. We first add a learnable modality token to explicitly indicate modality identity and reduce ambiguity when a shared backbone processes heterogeneous inputs:

$$\tilde{\mathbf{X}}^{\text{app}} = \mathbf{X}^{\text{app}} + \mathbf{1}\left(\mathbf{m}^{\text{app}}\right)^\top, \quad \tilde{\mathbf{X}}^{\text{geo}} = \mathbf{X}^{\text{geo}} + \mathbf{1}\left(\mathbf{m}^{\text{geo}}\right)^\top. \tag{3}$$

Here, $\mathbf{1} \in \mathbb{R}^{N \times 1}$ is an all-ones column vector used to broadcast the learnable modality token $\mathbf{m} \in \mathbb{R}^d$ to all $N$ tokens.

**Local Cross-Modality Attention**. Before the standard self-attention in each DiT block, we insert a lightweight local

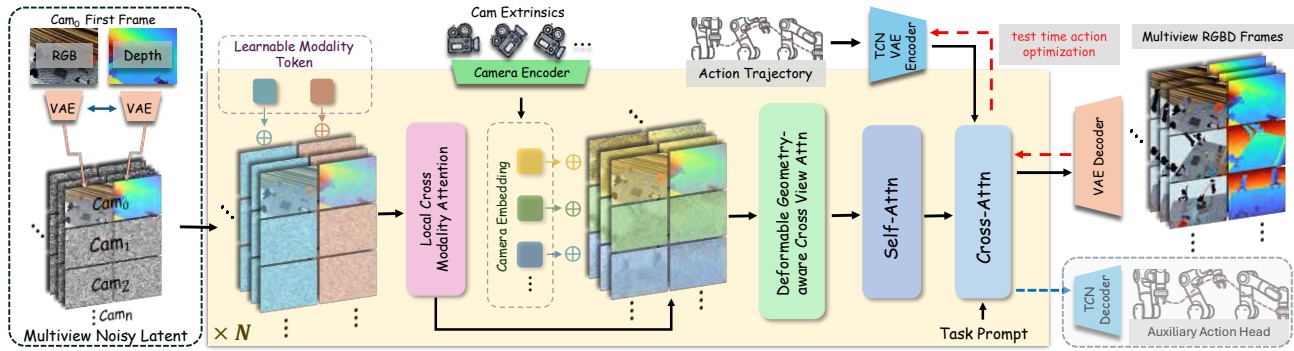

*Figure 1.* The overview of our main pipeline.

cross-modality attention module that exchanges complementary features only within a small neighborhood. For each position $i$, we define a local window $\mathcal{N}_r(i)$ on the 2D latent grid with radius $r$ on the geometry token grid and compute appearance to geometry attention as, and vise verse.

$$\mathbf{y}_i^{\mathrm{a \leftarrow g}} = \mathrm{Attn}\Big( \tilde{\mathbf{x}}_i^{\mathrm{app}} \mathbf{W}_Q^{\mathrm{app}}, \ \tilde{\mathbf{X}}_{\mathcal{N}_r(i)}^{\mathrm{geo}} \mathbf{W}_K^{\mathrm{geo}}, \ \tilde{\mathbf{X}}_{\mathcal{N}_r(i)}^{\mathrm{geo}} \mathbf{W}_V^{\mathrm{geo}} \Big). \tag{4}$$

We inject the retrieved feature with a gated residual update:

$$\hat{\mathbf{x}}_i^{\mathrm{app}} = \tilde{\mathbf{x}}_i^{\mathrm{app}} + \gamma_{\mathrm{app}} \mathbf{y}_i^{\mathrm{a \leftarrow g}}, \ \ \hat{\mathbf{x}}_i^{\mathrm{geo}} = \tilde{\mathbf{x}}_i^{\mathrm{geo}} + \gamma_{\mathrm{geo}} \mathbf{y}_i^{\mathrm{g \leftarrow a}}. \tag{5}$$

Here $\gamma^{\mathrm{app}}$ and $\gamma^{\mathrm{geo}}$ are channel-wise gates that amplify fusion when cross modality cues are reliable and suppress harmful transfers under heavy noise or imperfect alignment. The locality constraint provides a strong inductive bias for cross modality correspondence and reduces the cost of cross attention from global quadratic matching to $O(Nk)$.

### 4.2. Learning Geometric Consistency Across Views

**Camera Embedding as a View Token**. Similar to the modality token, we attach a view-specific token to each view. Instead of learning an extra embedding, we directly use the camera embedding as this discriminative token.

A common choice is to flatten the extrinsic matrix into a $3 \times 4$ vector and feed it to an MLP projector. However, this representation is sensitive to the global coordinate frame and entangles rotation and translation in a way that does not explicitly reflect the relative viewing configuration, which can hurt generalization across rigs and scenes. Instead, we parameterize each camera in spherical coordinates around a shared look-at point $\mathbf{p}$. We estimate $\mathbf{p}$ by least squares as the point closest to all camera optical axes:

$$\mathbf{p} = \arg \min_{\mathbf{x} \in \mathbb{R}^3} \sum_v \big\| \big( \mathbf{I}_{3 \times 3} - \mathbf{d}_v \mathbf{d}_v^\top \big) (\mathbf{x} - \mathbf{c}_v) \big\|_2^2, \tag{6}$$

where $\mathbf{c}_v$ is the camera center and $\mathbf{d}_v$ is the unit viewing direction. With $\mathbf{r}_v = \mathbf{c}_v - \mathbf{p}$ and $\rho_v = \|\mathbf{r}_v\|_2$, we compute yaw $\psi_v$ and pitch $\theta_v$ from $\mathbf{r}_v$, and take roll $\phi_v$ from the camera rotation. We then apply Fourier features with

$K = 2$ frequencies to $(\psi_v, \theta_v, \phi_v)$ and concatenate $log(\rho_v)$, yielding a compact 13-D embedding:

$$\mathbf{e}_v = \big[ \gamma(\psi_v), \gamma(\theta_v), \gamma(\phi_v), log(\rho_v) \big] \in \mathbb{R}^{13}. \tag{7}$$

Our representation makes scale cues easier for the model to access and exploit than flattened extrinsics, where scale is entangled with rotation and translation.

Furthermore, it also serves as a discriminative view token that allows the shared backbone to distinguish tokens from different viewpoints. Combined with our height-wise view concatenation, this design naturally supports a variable number of views by simply appending additional view streams along the height dimension.

**Geometry-aware Deformable Cross-view Attention**. We propose a lightweight, geometry-aware deformable cross-view attention to capture multi-view correspondences. Given known camera parameters, a token at view $v$ induces an epipolar line on any other view $u$, along which a small subset of tokens is highly correlated with the query. Instead of performing global cross-attention over the entire feature map, we sparsely sample $K$ candidate key-value locations along the corresponding epipolar line in each of the other $V - 1$ views, leading to only $(V - 1)K$ candidates per query. We then apply multi-head attention over this epipolar-restricted set to efficiently model cross-view correlations under geometric constraints.

Since latents have a coarse spatial resolution, we further introduce a deformable refinement to improve alignment. For each sampled location on the epipolar line, we predict a small offset using an MLP conditioned on the query feature, the initially sampled key feature, and their similarity, and clamp the magnitude by a maximum offset. The refined sampling location is thus adjusted from the geometry-prior epipolar position to a more discriminative and better-aligned position before attention aggregation. Formally, for a query token $\mathbf{q}i^v$ in view $v$ and a target view $u$, we uniformly sample $K$ locations $\{\mathbf{p}_{i,k}^{u,0}\}_{k=1}^K$ along the induced epipolar line, extract their features $\mathbf{f}_{i,k}^{u,0}$ from $\mathbf{X}^u$ via bilinear sampling.

We then predict the offset by:

$$\Delta \mathbf{p}_{i,k}^u = \mathrm{clip}\Big(\mathrm{MLP}_{\mathrm{off}}\big[\mathbf{q}_i^v, \mathbf{f}_{i,k}^{u,0}, s_{i,k}^u\big], \mathrm{max\_offset}\Big), \quad (8)$$

where $s_{i,k}^u$ is the cosine similarity between $\mathbf{q}_i^v$ and $\mathbf{f}_{i,k}^{u,0}$. Then the refined samples can be obtained by $\mathbf{p}_{i,k}^u = \mathbf{p}_{i,k}^{u,0} + \Delta\mathbf{p}_{i,k}^u$, which can be used to compute cross-view attention on the sparse set $\{\mathbf{p}_{i,k}^u\}$. Finally, the standard self-attention operates on the fused tokens to propagate information globally and enforce long-range consistency.

### 4.3. Trajectory Conditioning as a Style Code

Since we generate robotic manipulation scenes, the action trajectory is a key driver of the resulting motion. The trajectory is highly task-dependent, and we want the model to capture how the action sequence governs the motion in the generated video. A naive solution is to condition the generator on the per-step action sequence aligned to each video frame. In practice, this imposes a brittle one-to-one temporal alignment between action steps and frames, and it exposes high-frequency control details that are only weakly observable in pixels, which encourages shortcut learning and degrades generalization under rate mismatch or temporal resampling. Moreover, manipulation lie on a structured, low-dimensional trajectory space with smooth temporal correlations, so conditioning actions at the trajectory level is both more stable and less ill-posed than step-wise mappings.

To address this, we compress the action sequence into a low-dimensional manifold and use it as a *style code* injected into the generative model. We view a trajectory as a style signal because it specifies how the motion unfolds, including its temporal rhythm, smoothness, and coarse phase structure, while leaving appearance and fine visual details to other conditions. Concretely, we train a TCN-based VAE to encode an action sequence $\mathbf{a}_{1:T}$ into a latent token sequence $\mathbf{z} \in \mathbb{R}^{S \times C=32}$. we use $\mathbf{z}$ as $S$ style tokens and inject them into the generator via cross-attention.

$$\mathbf{z} = \mathrm{Enc}_{\mathrm{TCN}}(\mathbf{a}_{1:L}), \quad \hat{\mathbf{a}}_{1:L} = \mathrm{Dec}_{\mathrm{TCN}}(\mathbf{z}). \quad (9)$$

We train the VAE with the standard objective

$$\mathcal{L}_{\mathrm{VAE}} = \mathbb{E}_{q_\phi(\mathbf{z}|\mathbf{a})}\big[\|\mathbf{a} - \hat{\mathbf{a}}\|_2^2\big] + \beta\,\mathrm{KL}(q_\phi(\mathbf{z} \mid \mathbf{a}) \,\|\, p(\mathbf{z})), \quad (10)$$

and inject $\mathbf{z}$ into the video generator as a style code via cross-attention. To prevent the generator from selectively ignoring the trajectory condition, we add a lightweight *latent-consistency* head during training that reconstructs the conditioning latent tokens from the generator's final-layer representations (rather than predicting actions). Let $\mathbf{H}^{\mathrm{out}}$ denote the final-layer hidden tokens of the generator. We reconstruct the trajectory latents as $\hat{\mathbf{z}} = \mathrm{Proj}(\mathbf{H}^{\mathrm{out}}) \in \mathbb{R}^{S \times C}$, and impose a reconstruction loss against the conditioning latent $\mathbf{z}$: $\mathcal{L}_{\mathrm{traj}} = \|\hat{\mathbf{z}} - \mathbf{z}\|_2^2$. This auxiliary objective encourages the model to preserve trajectory-relevant information

throughout generation, the consistency head is only used for training and is discarded at inference.

### 4.4. Embodied Inference and Planning

At inference time, we are given a single-view scene observation and a text instruction $l$. Our objective is two-fold: (i) generate a plausible dynamic scene rollout that satisfies the instruction, and (ii) recover an executable action trajectory consistent with the generated rollout.

**Trajectory Latent Inference via Test-time Optimization.** Thanks to our trajectory conditioning, we can recover actions without using an additional action-prediction head for the generator. We first generate a dynamic rollout $\bar{\mathbf{V}}$ using text-only conditioning. We then freeze $\bar{\mathbf{V}}$ and optimize a randomly initialized trajectory latent $\mathbf{z}$ by backpropagation, searching for the conditioning latent that best reproduces the fixed rollout:

$$\mathbf{z}^\star = \arg\min_{\mathbf{z}}\ \mathcal{D}\big(G(l, \mathbf{z}), \bar{\mathbf{V}}\big) + \lambda\|\mathbf{z}\|_2^2, \quad (11)$$

where $G(\cdot)$ denotes the frozen generator conditioned on instruction $l$ and trajectory latent $\mathbf{z}$, and $\mathcal{D}(\cdot, \cdot)$ is a reconstruction metric (an $\ell_2$ distance). We then decode $\mathbf{z}^\star$ into an action sequence using the pretrained TCN decoder: $\hat{\mathbf{a}}_{1:T} = \mathrm{Dec}_{\mathrm{TCN}}(\mathbf{z}^\star)$. Optimizing in the learned low-dimensional trajectory latent space is typically more stable and yields smoother, more executable actions after decoding, compared to directly optimizing a full per-step action sequence as a conditioning signal.

**Residual Inverse Dynamics Module.** The action sequence recovered by optimization may still contain errors. Therefore, we introduce a residual inverse dynamics module that treats the decoded trajectory as a strong prior. Specifically, given two consecutive 3D point sets $\mathcal{P}_t$ and $\mathcal{P}_{t+1}$ together with the prior action $\mathbf{a}_t^{\mathrm{prior}}$ from our trajectory decoder, the residual inverse dynamics network predicts a correction $\Delta\mathbf{a}_t$ and outputs the refined action $\mathbf{a}_t = \mathbf{a}_t^{\mathrm{prior}} + \Delta\mathbf{a}_t$. This design offers two advantages. First, learning only the residual alleviates the ill-posedness of inverse dynamics (for example, multiple actions can explain similar transitions) by anchoring prediction around a plausible trajectory prior. Second, since $\mathbf{a}_t^{\mathrm{prior}}$ already encodes trajectory-level intent and temporal structure, the residual module can focus on local alignment and execution-level adjustment rather than reconstructing the entire action from scratch.

## 5. Experiments

We first evaluate 4D scene generation on both synthetic and real-world datasets and analyze the results in Sec. 5.1. We then conduct robotic manipulation experiments on two simulated platforms and one real-world setup to demonstrate how the predicted dynamic scenes support action planning in Sec. 5.2. More qualitative results and discussions are

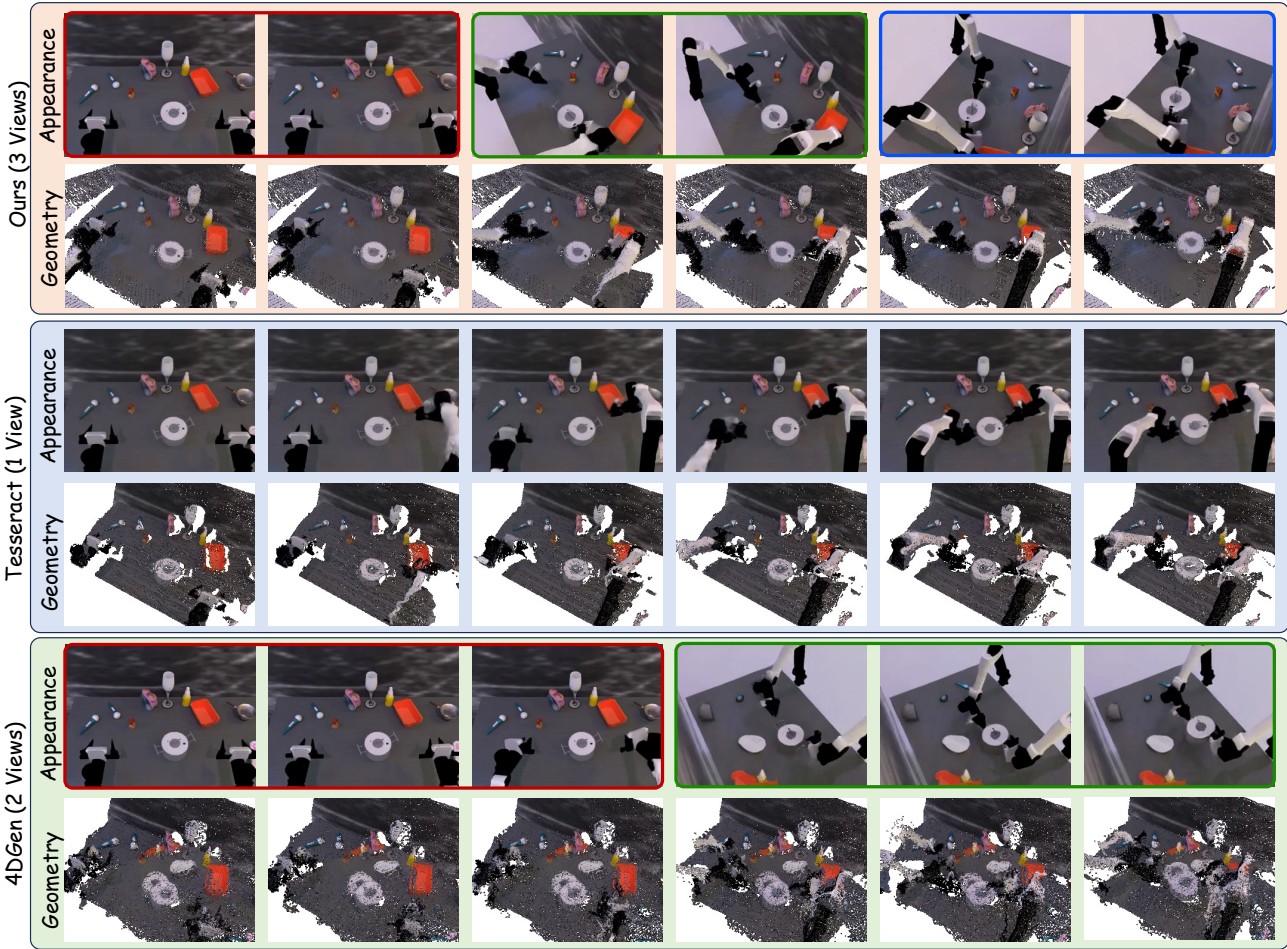

*Figure 2.* Qualitative Results of 4D Generation on RoboTwin dataset. Red, green, and blue boxes represent different viewpoints.

provided in our Supplemental Material and Appendix.

## 5.1. 4D Scene Generation

**Dataset**. Our experiments are conducted on two synthetic and one real-world dataset. For the synthetic data, we collect over 8,000 and 10,000 trajectories on RLBench (James et al., 2020) and RoboTwin2 (Chen et al., 2025a), each with 10 tasks. Each episode contains 16 RGB-D camera views with known camera parameters, arranged on a semi-sphere around the scene, along with a text instruction specifying the task target and the corresponding action sequence. For real-world evaluation, we deploy a robot arm platform equipped with 4 RGBD cameras to collect 14 manipulation tasks by tele-operation. We also collect corresponding actions. Details are provided in the Appendix.

**Metrics**. For 4D generation, we evaluate both appearance and geometry quality. For appearance, we report FVD, SSIM, and PSNR. For geometry, we report depth metrics including Absolute Relative Error (AbRel), RMSE, $\delta_1$, and 3D point metrics including Chamfer Distance (CD) and

Earth Mover Distance (EMD).

**Baselines**. We compare our methods with 3 manipulation world model baselines.

- UniPi (Du et al., 2023) is a video world model that generates 2D dynamic scenes and uses an image-based inverse dynamics model to map predictions to action chunks. Since UniPi does not predict geometry, we augment it with Depth Anything 3 (Lin et al., 2025) and obtain temporally continuous depth maps by optimizing the per-frame predictions with a depth alignment constraint to the first frame, which we refer to as UniPi*.
- TesserAct (Zhen et al., 2025) is a 4D world model that generates single-view RGB-DN sequences and employs a point-based IDM to derive executable actions.
- 4DGen (Liu et al., 2025) is a two-view 4D world model that generates Dust3R-style pointmaps and uses pose estimation to convert the predicted geometry into actions.

In addition, because our method uses WAN2.2 as its backbone, we implement all methods on a common WAN2.2

TI2V backbone where applicable.

**Implementation**. We build our method on WAN2.2 TI2V and train it with an SFT-style fine-tuning protocol. We use a randomized masking strategy: with probability 0.5 we randomly mask a variable number of frames in the source RGB-D video latents, and with probability 0.5 we provide only the first frame. At test time, we condition on the first frame only. Action sequences are encoded by a pre-trained TCN-VAE, which is frozen during training. We use a trajectory-conditioning dropout schedule: $p_{\mathrm{drop}} = 0$ for the first 50 epochs, then linearly increasing it to 0.5 for the remainder of training to encourage learning of the text-only marginal. We set $\lambda_1 = 0.1$ for $\mathcal{L}_{\mathrm{traj}}$.

**Variable-view Inference**. We support a variable number of views with two inference strategies: (1) We concatenate views along the height dimension and condition each view with its camera embedding. During training, we randomly sample 2–3 views per scene. Despite never seeing more views, this strategy can often extrapolate to 4–5 views at test time by simply appending additional view streams. This suggests promising scalability. (2) We first generate a subset of views, then complete extra views in a second sampling run by freezing the already-generated latent regions and denoising only the missing-view regions. We find masked completion yields more stable and higher-quality results, and thus use it by default in all experiments. We use mode 2 by default in all main experiments unless otherwise specified. Since prior baselines (e.g., 4DGen under the DUSt3R paradigm) are maximally limited to two views, we adopt three-view in all comparisons in the main paper and provide more-view examples and an analysis of both modes in the Appendix.

**More Consistent 4D Geometry Prediction**. Our model predicts view-consistent RGB-D sequences for a variable number of camera views. Back-projecting and fusing the multi-view predictions yields more complete 3D point trajectories with fewer occlusion-induced holes. Quantitative results in Table 1 show improved depth and point-cloud metrics, indicating stronger cross-view geometric consistency than prior methods. Fig. 2 further provides qualitative evidence, where our predictions fuse into a more complete point cloud with fewer misalignments and missing regions. Fig. 3 shows similar fused trajectories in the real-robot setting. More qualitative results are included in the Appendix and supplemental videos.

**High-quality Appearance Generation**. We expect that using more views can further improve geometry metrics by increasing surface coverage and reducing occlusions, at the cost of higher computation. Notably, our method also achieves better appearance quality. We attribute this to the fact that stronger geometry provides more reliable structural cues for rendering consistent appearance, and our explicit

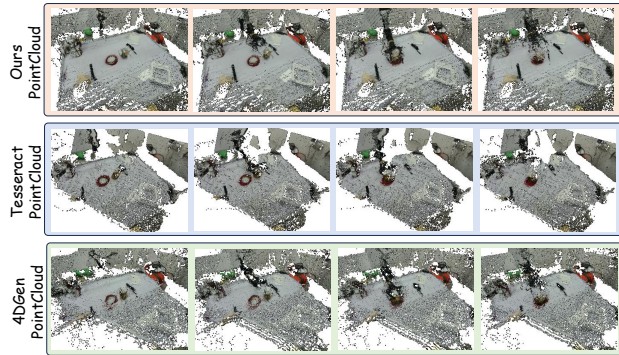

*Figure 3*. Generated Geometries on Real-World Robot dataset

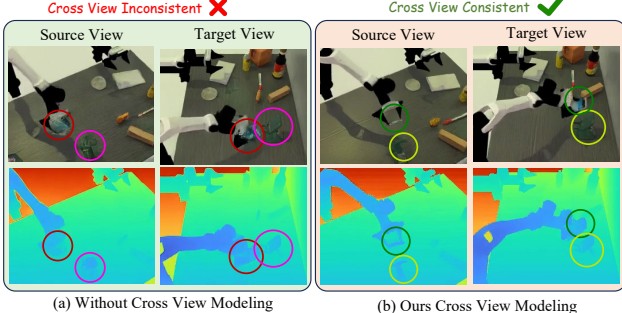

(a) Without Cross View Modeling    (b) Ours Cross View Modeling

*Figure 4*. Effect of geometry-aware cross view modeling

cross-modality fusion encourages effective information exchange between RGB and depth.

**Ablations on Cross View Modeling**. We explicitly fuse features across views. On RoboTwin (Table 2), we compare our Geometry-aware Deformable Cross-View Attention (Ours) with removing cross-view fusion (w/o view) and replacing it with a simple epipolar attention (EA). Both alternatives consistently degrade performance. As illustrated in Fig. 4, the drop mainly stems from increased cross-view inconsistency, which leads to misaligned geometry and less coherent multi-view predictions—for example, objects shift across views and depth boundaries no longer agree. In contrast, our cross-view modeling better preserves consistent geometry and occlusion ordering.

**Analysis on Cross-Modality Modeling**. Table 2 also reports results without modality fusion (w/o Mod), highlighting the importance of explicit modality alignment. We further provide qualitative comparisons in Fig. 5 to illustrate this effect. Without cross-modality modeling, the predicted RGB often drifts from the depth and fails to stay aligned with object boundaries across views and time. While our model produces consistently aligned RGB-D predictions, indicating more reliable cross-modality correspondence.

## 5.2. Embodied Action Planning

**Dataset and Metrics**. We further evaluate manipulation performance on all tasks from the previous section, covering

*Table 1.* 4D generation results on the RLBench, RoboTwin, and our real-world datasets. The SSIM, AbRel, RMSE, CD, and EMD are scaled by $10^2$.

| Method | Appearance | | | Depth | | | Point Cloud | |
|---|---|---|---|---|---|---|---|---|
| | PSNR↑ | SSIM↑ | FVD↓ | AbRel↓ | RMSE↓ | $\delta_1$ ↑ | CD↓ | EMD↓ |
| *RLBench* | | | | | | | | |
| UniPi* | **23.88** | 91.7 | 19.94 | 117.9 | 43.2 | 91.6 | 15.0 | 20.6 |
| 4DGen | 22.25 | 87.1 | 20.51 | 91.8 | 29.3 | 94.1 | 10.9 | 16.0 |
| TesserAct | 23.86 | **92.8** | 27.77 | 91.8 | 29.4 | 96.8 | 11.0 | 16.3 |
| Ours | 23.31 | 90.8 | **18.57** | 90.5 | **29.1** | **97.1** | **9.6** | **15.3** |
| *RoboTwin* | | | | | | | | |
| UniPi* | **22.98** | 89.2 | 22.18 | 5.52 | 18.13 | 95.1 | 9.88 | 20.53 |
| 4DGen | 22.18 | 85.2 | 24.61 | 3.00 | 13.90 | 96.6 | 7.18 | 10.62 |
| TesserAct | 22.65 | 89.8 | 27.29 | 3.71 | 15.07 | 97.3 | 7.11 | 10.28 |
| Ours | 22.91 | **90.2** | **21.93** | **2.60** | **12.30** | **97.4** | **6.51** | **9.90** |
| *Real-world Dataset* | | | | | | | | |
| UniPi* | **22.53** | 90.62 | 28.62 | 39.95 | 42.69 | 67.55 | 58.41 | 63.22 |
| 4DGen | 21.34 | 89.75 | 25.60 | 23.36 | 29.61 | 79.59 | 17.32 | 15.61 |
| TesserAct | 22.27 | **91.50** | 50.79 | 30.56 | 33.17 | 34.16 | 38.47 | 34.65 |
| Ours | 21.82 | 89.98 | **23.08** | **20.79** | **25.11** | **82.18** | **13.06** | **14.37** |

*Table 2.* Ablation studies on cross- view and modality modules

| Method | Appearance | | | Depth | | | Point Cloud | |
|---|---|---|---|---|---|---|---|---|
| | PSNR↑ | SSIM↑ | FVD↓ | AbRel↓ | RMSE↓ | $\delta_1$↑ | CD↓ | EMD↓ |
| w/o view | 21.47 | 88.3 | 24.85 | 3.34 | 14.30 | 95.7 | 8.34 | 17.50 |
| EA | 22.43 | 89.2 | 23.36 | 3.03 | 12.70 | 96.9 | 7.33 | 12.50 |
| w/o mod | 20.16 | 83.8 | 25.25 | 4.03 | 16.77 | 95.2 | 7.51 | 16.80 |
| Ours | **22.91** | **90.2** | **21.93** | **2.60** | **12.30** | **97.4** | **6.51** | **9.90** |

both synthetic and real-world settings. The success rate over 100 episodes is adopted for the Metrics.

**Baselines**. We include all methods from Sec. 5.1 in our manipulation evaluation, each with its own method to convert 4D generation to action. UniPi uses an image-based inverse dynamics model to map predicted observations to action chunks. 4DGen converts generated geometry into object or gripper poses using the FoundationPose. TesserAct applies a 3D point-based inverse dynamics module to predict actions. We also include a point-based ACT trained with imitation learning as an additional baseline.

**Implementation**. We first run text-only inference to predict a 4D scene and keep the prediction fixed. We then initialize a random trajectory latent and optimize it for 100 backprop-agation steps so that the trajectory-conditioned generation matches the fixed 4D prediction. The TCN decoder decodes the optimized latent into an executable action sequence, which is further refined by the residual IDM (Sec. 4.4) to produce the final trajectory.

**Results and Analysis**. The results in Table 3 show that our method consistently outperforms all baselines across tasks and platforms. We attribute this to two factors. First, multi-view generation yields more complete geometry, providing stronger cues for action inference, while single- or two-view methods are more brittle under occlusion. Second, trajectory-level latent optimization directly searches for a

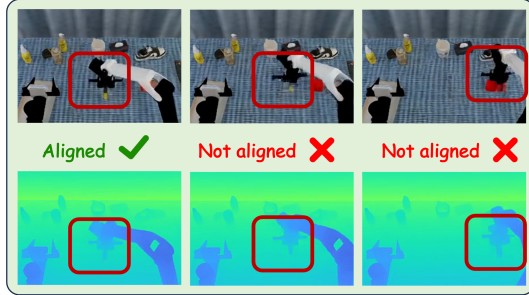

(a) Without cross modality modeling

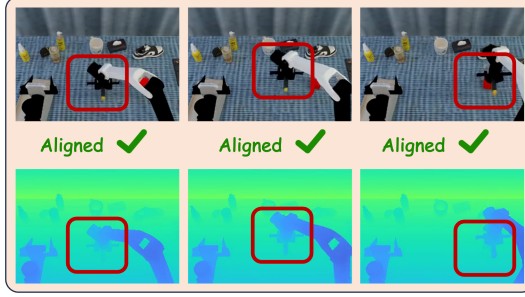

(b) Ours

*Figure 5.* Effect of explicit cross-modality modeling

conditioning trajectory that best explains the imagined 4D future, producing actions that better match the generated dynamics. Moreover, our residual IDM only predicts a correction to this strong prior, reducing ambiguity compared to solving inverse dynamics from scratch. Even in a real robot platform (Table 4), our model still shows superior performance than TesserAct.

*Table 3.* Manipulation results on RLBench and RoboTwin.

| Dataset | Baselines | | | | Ablations | | | |
|---|---|---|---|---|---|---|---|---|
| | P-ACT | UniPi* | 4DGen | TesserAct | Act Head | full IDM | w/o R-IDM | full model |
| RLBench | 60.4 | 34.6 | 47.0 | 67.3 | 72.5 | 68.8 | 69.0 | **72.6** |
| RoboTwin | 20.5 | 16.3 | 40.2 | 33.9 | 42.5 | 41.7 | 42.8 | **43.0** |

*Table 4.* Success rates for some tasks on the real robot platform.

| Method | Arrange Boxes | Cap Bottle | Open Drawer | Place Fruits | Put Orange | Stack Cubes |
|---|---|---|---|---|---|---|
| TesserAct | 7 | 27 | 37 | 17 | **66** | 45 |
| Ours | **15** | **33** | **56** | **23** | 63 | **50** |

**Effect of Trajectory Latent Optimization**. We evaluate the importance of test-time optimization over the trajectory latent. As a comparison, we construct a variant that skips latent optimization and directly uses the action-head prediction as the trajectory prior for the residual IDM. We denote this baseline as *Act-Head* in Table 3, which indicates consistent benefit from test-time latent optimization.

**Ablation of Residual IDM**. We further analyze the role of the residual IDM with two additional variants. First, following (Zhen et al., 2025), we train a point-based inverse

*Table 5.* Manipulation ablations on fusion modules and trajectory representation. All variants use the same 3-view RGB-D input.

| Dataset | w/o view | w/o mod | w/o view+mod | cat depth | w/o TCN-VAE | Full |
|---|---|---|---|---|---|---|
| RLBench | 68.8 | 67.2 | 66.5 | 66.1 | 68.1 | **72.6** |
| RoboTwin | 38.0 | 38.6 | 36.2 | 39.4 | 41.5 | **43.0** |

*Table 6.* Efficiency comparison under the same video backbone on a single NVIDIA RTX H200. Time is reported as generation + post-processing.

| Benchmark | 4DGen | TesserAct | Ours | +ActionInit |
|---|---|---|---|---|
| RLBench SR | 47.0% | 67.3% | 72.6% | **76.5%** |
| RLBench Time | $51 + 22 = 73s$ | $37 + 16 = 53s$ | $42 + 16 = 58s$ | $42 + 5 = 47s$ |
| RoboTwin SR | 40.2% | 33.9% | 43.0% | **46.6%** |
| RoboTwin Time | $83 + 25 = 108s$ | $68 + 23 = 91s$ | $75 + 23 = 98s$ | $75 + 9 = 84s$ |

dynamics model and feed the generated 4D scenes into it to predict action chunks; we refer to this baseline as *Full IDM*. Second, we remove the residual IDM and directly execute the action sequence decoded from the optimized trajectory latent (*w/o R-IDM*). Results are reported in Table 3. This shows that *w/o R-IDM* and *Full IDM* both underperform our full model on RLBench, validating the effectiveness of residual refinement.

**Manipulation Ablations on Fusion and Trajectory Representation**. We further extend the fusion ablations to downstream manipulation. As shown in Table 5, removing cross-view fusion, cross-modality fusion, or both consistently degrades success rates under the same 3-view RGB-D input. The *cat depth* baseline directly concatenates depth with RGB, but performs worse because the pretrained video backbone is optimized for 3-channel RGB inputs and RGB-D tokens receive distant positional embeddings after flattening. In addition, removing the TCN-VAE also hurts performance, confirming the benefit of a compact trajectory latent for action inference.

**Auxiliary Action Initialization and Inference latency**. We further compare the inference latency of different action-inference pipelines under the same video backbone on a single NVIDIA RTX H200 in an offline setting. Since the generation backbone is shared, the latency difference mainly comes from post-processing: 4DGen maps generated points to robot poses, TesserAct optimizes temporal alignment, while our method optimizes the trajectory latent. As shown in Table 6, our method achieves strong performance with comparable inference time.

Moreover, we can initialize the trajectory latent with the action-head prediction. This provides a better starting point and reduces the required optimization to only 10–15 steps. We denote this variant as *+ActionInit*. It further improves success rates from 72.6% to 76.5% on RLBench and from 43.0% to 46.6% on RoboTwin, while reducing total inference time from 58s to 47s and from 98s to 84s, respectively. These results show that trajectory-latent optimization can

be beneficial from this extra surprise, which is because the initialization will not introduce any additional training modules.

# 6. Conclusion

We propose an embodied multi-view 4D world model that jointly supports future prediction and action inference, with explicit cross-modality and cross-view fusion for geometry-consistent RGBD reasoning. To address the ill-posedness of inverse dynamics, we introduce a test-time action inference scheme that leverages a trajectory prior refined by a residual inverse dynamics model for reliable execution. We evaluate on RoboTwin, RLBench, and a newly collected real-robot 4D multiview dataset with 14 tasks, spanning 34 manipulation tasks in total, where our method consistently outperforms strong baselines in both 4D scene generation and downstream manipulation. Extensive analyses validate the key design choices and provide practical insights for future embodied world modeling, suggesting that view-consistent 4D prediction can serve as a useful geometric interface between imagined futures and executable robot actions.

# Acknowledgements

This work was partially supported by the National Natural Science Foundation of China (No. 62306261), HK RGC-Early Career Scheme (No. 24211525), ITSP Platform Project (No. ITS/600/24FP), the SHIAE Grant (No. 8115074), Hong Kong RGC Strategic Topics Grant (No. STG1/E-403/24-N), and CUHK-CUHK(SZ)-GDST Joint Collaboration Fund (No. YSP26-4760949). This work was also supported in part by the Centre for Perceptual and Interactive Intelligence, a CUHK-led InnoCentre, and the AI Chip Center for Emerging Smart Systems (ACCESS), both under the InnoHK initiative of the Innovation and Technology Commission of the Hong Kong Special Administrative Region Government.

# Impact Statement

Our method can improve the reliability of embodied manipulation by enabling geometry-consistent 4D prediction and test-time action inference, which is especially beneficial in cluttered and occluded environments. We also find encouraging scale-up behavior: generating more views tends to yield more complete and better-aligned 4D reconstructions, suggesting a practical path to stronger performance with increased compute or view budgets. Potential risks include misuse for producing realistic synthetic visual data or misleading reconstructions, we therefore advocate responsible release and safety-aware deployment.

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

# A. Implementation Details

**Training Strategy** We train a separate model for each dataset. For tasks with different manipulated objects, we keep a fixed instruction template and only replace the object noun, for example, "pick up the coke bottle" versus "pick up the can".

The diffusion input is a noise latent of shape $B \times T \times C \times h \times w$. With probability $0.5$, we replace the corresponding noise region with only the first frame of the first view; with probability $0.5$, we replace it with the full video latent from the first view. When the full video is provided, we further randomly mask an arbitrary number of frames in its latent. This design exposes the model to inputs of varying information density, enabling it to generate 4D dynamics from a single-frame observation, and to complete missing timesteps while remaining highly consistent with the given frames.

In addition to text, we condition the model on the trajectory latent obtained by encoding the action sequence with a TCN-VAE. During early training, we always provide the trajectory latent so the model learns the joint distribution over text, observations, and actions. In later training, we gradually increase the probability of masking the trajectory latent and replacing it with a null trajectory token. This null token serves as a placeholder that can also be optimized by backpropagation at test time. As a result, the model learns to generate dynamic scenes from text alone while retaining the ability to leverage action conditioning when available. Overall, this training strategy supports flexible conditioning with different observation densities and both action-conditioned and action-free generation.

**Diffusion Model Details** Our diffusion backbone is based on the WAN2.2 TI2V 5B DiT model. We train with a learning rate of $10^{-5}$ using a 1000-step linear warmup followed by a constant schedule. We use AdamW with $\epsilon = 10^{-8}$ and weight decay $0.01$. We release all the original blocks for finetunin,g and all additional blocks are initialized as zero weights and bias for protect the prior knowledge in WAN2.2.

**Local Cross Modality Attention** We implement a lightweight local cross-attention module to exchange information between two aligned latent feature maps (appearance and geometry) before the standard self-attention in each DiT block. After width-wise concatenation within each view, we split the fused tensor into two branches (RGB/appearance and depth/geometry), add a learnable modality embedding to disambiguate the two streams, and perform symmetric local fusion by applying cross-attention in both directions (geo→app and app→geo) with residual updates. Concretely, given a query map and a key-value map of shape $(B, T, H, W, C)$, we first apply two independent linear "pre-projections" to reduce the channel dimension to a smaller hidden size, then form multi-head $Q, K, V$ in this hidden space with RMSNorm applied to $Q$ and $K$. For each spatial location, attention is restricted to a fixed $k \times k$ neighborhood (optionally with dilation) centered at the corresponding position on the key-value map. We extract all local windows efficiently using `unfold` (with zero padding) and build a padding mask by detecting all-zero padded keys, so boundary tokens do not attend to invalid locations. Attention is computed either via fused scaled dot-product attention for speed, or via an explicit softmax path, and the output is projected back to the original input dimension while preserving the original tensor shape. This yields an efficient $O(HWk^2)$ local cross-modality fusion that provides a strong inductive bias for correspondence while avoiding global quadratic matching.

**Geometry-Aware Deformable Cross View Attention** We implement cross-view fusion via a sparse, geometry-grounded attention module that operates on multi-view latent features $x \in \mathbb{R}^{B \times V \times C \times H \times W}$. In each DiT block, we reshape the token grid into an explicit view dimension and apply the cross-view module before global self-attention, then add the result back to the fused tokens via a residual connection. Given camera extrinsics and the latent grid resolution, we precompute an epipolar sampling basis (cached across calls) and, for each query token in view $v$, gather $K$ candidate samples along the induced epipolar line in every other view using bilinear `grid_sample`, resulting in a sparse set of $(V-1)K$ keys/values per query. To compensate for coarse latent resolution and imperfect discretization, we further predict a small 2D offset for each sampled point using an MLP conditioned on the query feature, the initially sampled key feature, and their similarity, and clamp the offset magnitude by a maximum offset. We then re-sample keys/values at the refined positions and compute multi-head dot-product attention over the refined sparse set. Finally, the aggregated features are reshaped back to $(B, V, C, H, W)$ and fed to subsequent global self-attention layers to propagate information and enforce long-range consistency.

**Residual Inverse Dynamics Model** We implement the residual inverse dynamics module as a lightweight point-based network built upon a PointNet-style encoder. At each timestep, we first extract the interaction region in the workspace by cropping a fixed 2D window centered on the operational area (i.e., the middle workspace where the gripper and target typically appear) from the predicted RGB-D observations. We then back-project the cropped depth maps to 3D using known camera parameters and obtain two consecutive point sets $\mathbf{P}_t$ and $\mathbf{P}_{t+1}$ in a shared coordinate frame. To reduce background clutter and keep a fixed input size, we apply farthest point sampling (FPS) to subsample each point set to 8,192 points.

The two point clouds are concatenate and fed into a PointNet encoder with shared MLP layers and global max pooling to produce a compact transition feature. Conditioned on the trajectory prior action $a_t^{\text{prior}}$ decoded from the optimized trajectory latent, the R-IDM predicts a residual correction $\Delta a_t$ and outputs the refined action $a_t = a_t^{\text{prior}} + \Delta a_t$. During training, we supervise $\Delta a_t$ with the residual target $\Delta a_t^\star = a_t^\star - a_t^{\text{prior}}$, where $a_t^\star$ denotes the ground-truth action, using an $\ell_2$ regression loss. This residual formulation keeps the inverse dynamics learning well-conditioned (since the prior already provides trajectory-level intent and temporal structure) and lets the network focus on local execution-level adjustment from geometry changes rather than recovering the entire action from scratch.

## B. Simulation Setup

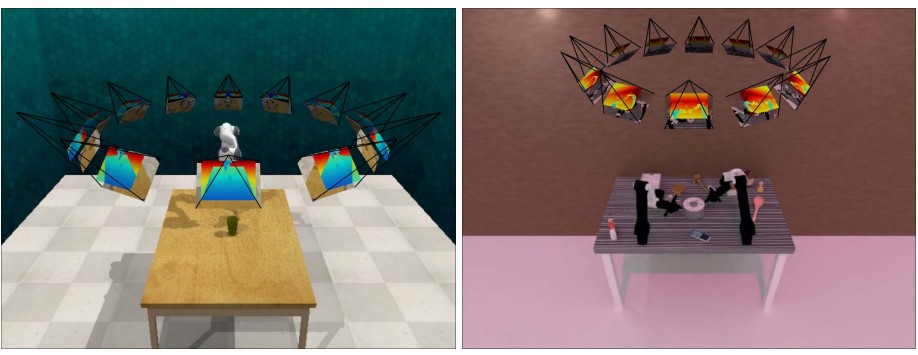

*(a)* RLBench camera layout      *(b)* RoboTwin camera layout

*Figure 6.* Simulator camera layout

**Camera Layout** We place 12 cameras uniformly around the scene. Each camera's horizontal-plane projection is separated by $30°$ and oriented toward the center of the robot worktable (see Fig. 6).

**Training View Selection** During training, we randomly sample 3 viewpoints and require that, for each sampled viewpoint, at least one camera lies within an angular separation of $90°$.

**Camera Settings** All cameras use a resolution of $320 \times 240$. For RLBench, the camera FOV is $40°$; for RoboTwin, the FOVY is $37°$.

## C. Real-World Robot Dataset Collection Setup and Demonstration

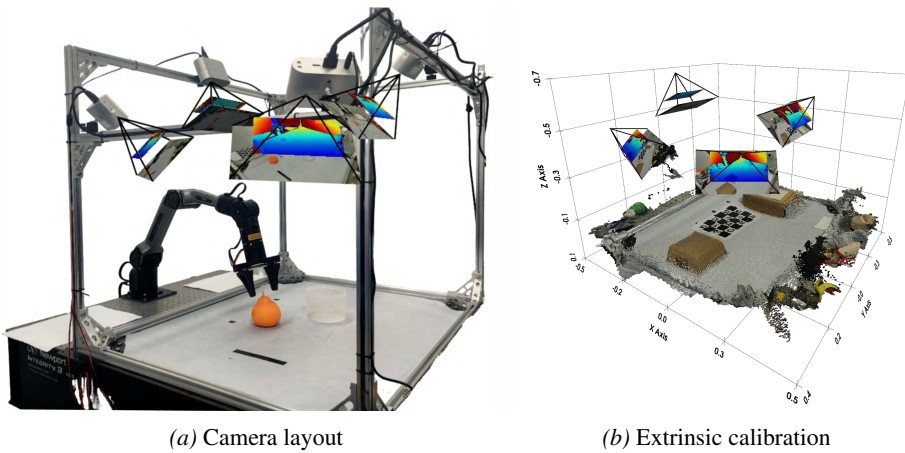

*(a)* Camera layout      *(b)* Extrinsic calibration

*Figure 7.* Real robot setup

**Hardware Setup and Calibration.** Our system consists of four Orbbec Femto Bolt RGB-D ToF cameras and two AgileX Piper robotic arms (see Fig. 7a). Data are collected on a workstation with an Intel i9-14900K CPU and an NVIDIA RTX 4090 GPU. We estimate camera extrinsics in two stages (Fig. 7b):

- **Coarse calibration.** Each camera observes a ChArUco board at the maximum resolution ($3840 \times 2160$) to obtain an initial estimate of the extrinsics.
- **Refinement.** We back-project RGB-D frames into point clouds and perform ICP on cropped overlapping regions. Fixing one camera as the reference, we sequentially refine the extrinsics of the remaining cameras.

**Teleoperation Setup** We use a leader–follower configuration: one arm serves as the teleoperation leader, while the other arm acts as the follower for data collection. The follower mirrors the leader's joint positions via direct joint-space mapping at a control frequency of 200 Hz.

**Data Frame Synchronization** Each camera records synchronized RGB-D frames at 15 FPS and $1280 \times 720$ resolution. Software timestamps are used to align camera and robot streams.

**Data Logging and Post-processing** To reduce CPU overhead during acquisition, raw streams are dumped directly to `.pkl` files. After collection, the data are converted to storage-efficient formats for downstream processing, and the resolution is down-sampled to $320 \times 180$.

**Manipulation Tasks** Fig. 8 summarizes each task's working environment and job description in our dataset. To increase task difficulty and scene diversity, we introduce randomly placed, task-irrelevant distractor objects in every scene. Moreover, for each episode of every task, the target object's pose is independently randomized, yielding a broad spatial distribution.

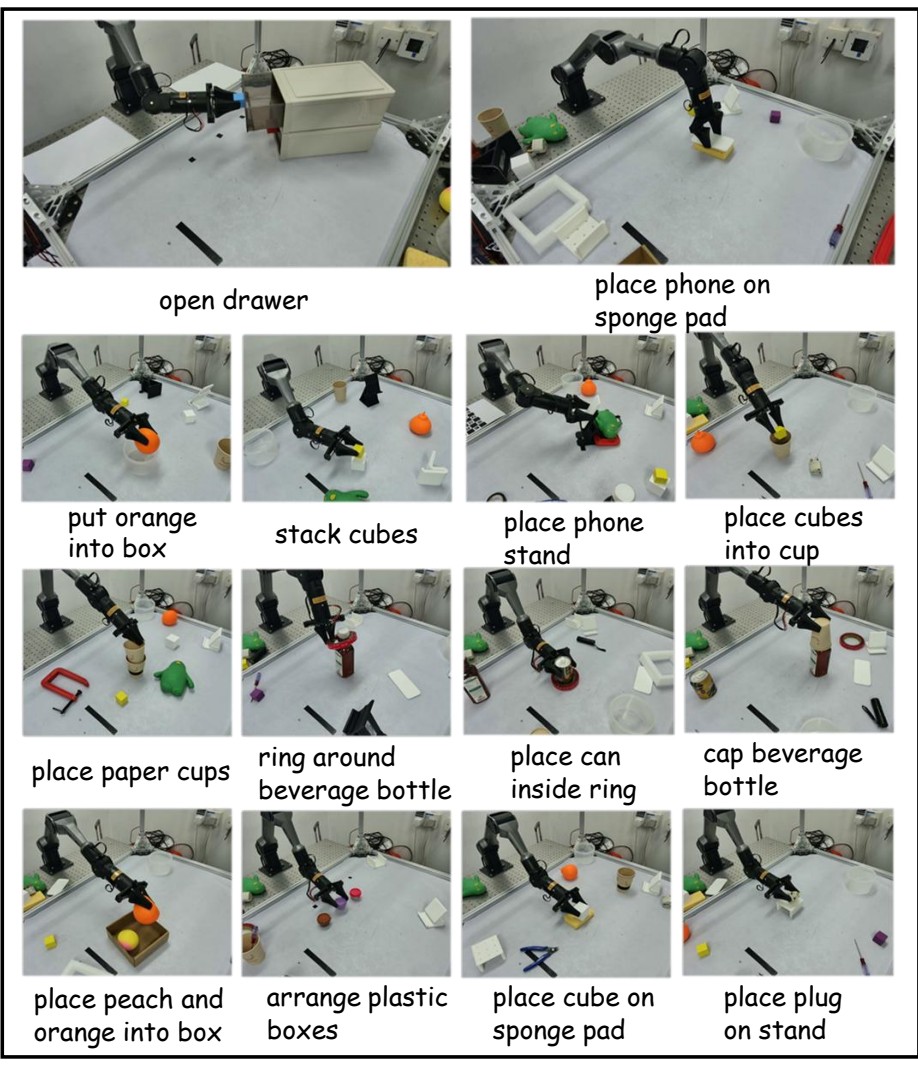

*Figure 8.* Illustrations of Manipulation Tasks in our real-world dataset

*Table 7.* Comparisons of 4D generation about our two modes strategies for supporting various multiview generations. SSIM, AbsRel, RMSE, $\delta_1$, CD, and EMD are scaled by $10^2$.

| Method | Appearance | | | Depth | | | Point Cloud | | Method | Appearance | | | Depth | | | Point Cloud | |
|---|---|---|---|---|---|---|---|---|---|---|---|---|---|---|---|---|---|
| | PSNR↑ | SSIM↑ | FVD↓ | AbsRel↓ | RMSE↓ | $\delta_1$↑ | CD↓ | EMD↓ | | PSNR↑ | SSIM↑ | FVD↓ | AbsRel↓ | RMSE↓ | $\delta_1$↑ | CD↓ | EMD↓ |
| 4DGen | 22.25 | 87.1 | 20.51 | 91.8 | 29.3 | 94.1 | 10.9 | 16.0 | 4DGen | 21.28 | 85.2 | 24.61 | 3.0 | 13.9 | 96.6 | 7.2 | 10.6 |
| TesserAct | 23.86 | 92.8 | 27.29 | 91.8 | 29.4 | 96.8 | 11.0 | 16.3 | TesserAct | 22.65 | 89.8 | 27.29 | 3.7 | 15.1 | 97.3 | 7.1 | 10.3 |
| Ours-mode1 | 22.55 | 89.3 | 21.67 | 90.8 | 29.2 | 96.9 | 10.2 | 15.2 | Ours-mode1 | 21.59 | 87.8 | 23.65 | 2.9 | 13.6 | 96.9 | 7.0 | 10.17 |
| Ours-mode2 | 23.31 | 90.8 | 18.57 | 90.5 | 29.1 | 97.1 | 9.6 | 15.3 | Ours-mode2 | 22.91 | 90.2 | 21.93 | 2.6 | 12.3 | 97.4 | 6.5 | 9.9 |

**Action Planning.** We evaluate success rates on 6 tasks. For each trial, we first verify whether the predicted video completes the task; incorrect completions are counted as failures. For correct videos, we decode actions with an inverse-dynamics model to align frames with robot actions, temporally interpolate the actions for smooth motion, and execute them to compute the final success rate. Results are reported in Table 4.

*Our method.* We randomly choose one input view and two target views. The decoded multi-view RGB-D video is projected and fused into a point-cloud sequence, then mapped to actions by the inverse-dynamics model, interpolated, and executed.

*TesserAct.* We randomly choose a single view and use its RGB-D frame as input; surface normals computed from depth form the RGB-DN input. The generated RGB-DN video is projected into a point-cloud sequence, then mapped to actions by the inverse-dynamics model, interpolated, and executed.

## D. Analysis of two Modes Multiview Inference

We support generating RGB-D videos with a variable number of camera views using two inference strategies. **Mode-1** predicts all views in a single sampling run by concatenating view latents along the height dimension, where each view is conditioned by its camera embedding. During training we randomly sample 2–3 views per scene; at test time, Mode-1 can often extrapolate to 4–5 views by appending additional view streams. **Mode-2** performs a two-pass masked-completion procedure: we first generate a subset of reference views, then complete additional views in a second sampling run by freezing the already-generated latent regions and denoising only the missing-view regions. We find masked completion yields more stable and higher-quality results, and thus use Mode-2 as the default setting for all main-paper experiments unless otherwise specified. Since prior baselines are typically limited to two views maximally, we adopt a three-view setup for all main-paper comparisons, and provide more-view examples and a detailed analysis of both strategies in Appendix.

We further provide two qualitative visualizations to complement the quantitative results in Fig. 11 and 12. Fig. 11 shows Mode-1 results on RoboTwin, and Fig. 12 shows Mode-2 results on RLBench. In each case, we compare our predictions against 4DGen under the same scene and camera setup. The visual comparisons highlight that our multi-view generation produces more coherent geometry across viewpoints, with fewer view-dependent artifacts and less cross-view drift. In particular, 4DGen often exhibits inconsistencies in object placement and depth discontinuities across views, which lead to misaligned fused point clouds, whereas our method preserves consistent occlusion ordering and produces sharper, more stable depth boundaries. These qualitative results align with the improvements in cross-view geometry metrics reported in Table 7.

## E. Additional Task-Level Details

### E.1. Per-Task Success Rates on RLBench

In the main paper, we report the overall average manipulation success rate of each method on different robotic platforms. To help readers gain a more fine-grained understanding of the results, we additionally provide per-task success rates in this appendix (Table 8). As shown in the table, our method consistently delivers the strongest performance across a broad range of task types. It achieves clear gains on relatively coarse, long-horizon interactions such as Close Drawer and Close Microwave (91 and 75), and remains highly robust in cluttered or occluded settings such as Open Drawer (98). Moreover, for tasks that require precise small-object handling or contact-sensitive control, e.g., Pick Up Cup, Push Button, and Play Jenga, our method still leads by a large margin (62, 89, and 97). Overall, these per-task results corroborate the aggregate numbers in the main paper and highlight the broad robustness of our approach across diverse manipulation scenarios.

*Table 8.* Detailed Comparisons of per manipulation task success rate on RLBench platform

| Method | Unplug Charger | Close Drawer | Close Grill | Close Microwave | Lamp On | Open Drawer | Pick Up Cup | Play Jenga | Push Button | Take Umbrella |
|---|---|---|---|---|---|---|---|---|---|---|
| P-ACT | 6 | 88 | **96** | 64 | 8 | 80 | 12 | 96 | 72 | **52** |
| UniPi* | 9 | 62 | 70 | 45 | 0 | 68 | 0 | 78 | 12 | 2 |
| 4DGen | 46 | 18 | 91 | 65 | 25 | 15 | 26 | 96 | 77 | 10 |
| TesserAct | 50 | 83 | 83 | 68 | 30 | 92 | 61 | 87 | 83 | 36 |
| Ours | **55** | **91** | **96** | **75** | **37** | **98** | **62** | **97** | **89** | 43 |

## E.2. Per-Task Success Rates on RoboTwin

To provide a more fine-grained view beyond the averaged success rates reported in the main paper, we summarize per-task success rates on RoboTwin in Table 9. Overall, our method achieves the best performance on a majority of tasks and shows particularly strong gains on contact-rich, long-horizon interactions. For example, we obtain the highest success rates on Adjust Bottle (69), Beat Hammer (42), Click Bell (38), Grab Roller (68), Lift Pot (42), and Place Container (72), consistently outperforming prior world-model baselines. While some tasks remain challenging (e.g., Object Stand and Phone Stand), the per-task breakdown highlights the broad robustness of our approach across diverse manipulation behaviors and confirms that the improvements are not driven by a small subset of tasks.

*Table 9.* Detailed comparisons of per manipulation task success rate on RoboTwin platform.

| Method | Adjust Bottle | Beat Hammer | Click Alarm clock | Click Bell | Grab Roller | Lift Pot | Move Pill bottle | Place Container | Object Stand | Phone Stand |
|---|---|---|---|---|---|---|---|---|---|---|
| P-ACT | 56 | 3 | 37 | 22 | 26 | 3 | 0 | 43 | 15 | 0 |
| UniPi* | 28 | 15 | 10 | 5 | 51 | 18 | 0 | 26 | 10 | 0 |
| 4DGen | 65 | 37 | 29 | 35 | 61 | 40 | **49** | 59 | 22 | 5 |
| TesserAct | 62 | 41 | **38** | 25 | 35 | 26 | 28 | 45 | **37** | 2 |
| Ours | **69** | **42** | 31 | **38** | **68** | **42** | 46 | **72** | 15 | **7** |

## F. Ablations on the novel camera embedding

Compared with the naive camera representation that flattens extrinsics (*Flatten Cam*), our spherical camera embedding yields consistently better geometric quality and cross-view consistency. As shown in Table 10, on RoboTwin our embedding improves depth accuracy and substantially reduces fused point-cloud errors, indicating more reliable multi-view fusion and fewer geometry misalignments; *Flatten Cam* achieves slightly lower FVD, but at the cost of notably worse geometry. On the real-world dataset, while *Flatten Cam* attains marginally higher PSNR and lower FVD, our embedding still leads to clearly better geometry metrics and point-cloud consistency, suggesting improved robustness to unseen camera rigs and real-world sensing noise. Overall, these results validate that the proposed camera embedding serves as an effective view-distinguishable token and is crucial for learning geometry-consistent multi-view 4D generation.

## G. Effect of the number of views on manipulation tasks

Table 11 studies the trade-off between performance and computation as we increase the number of views. Moving from 1 to 3 views yields a clear gain on RLBench (68.6→72.6), while further increasing to 4–5 views brings only marginal improvements (72.6→72.9/73.1). In contrast, the time cost grows steadily with more views; taking 3 views as the reference (time cost = 1.00), using 4 and 5 views increases the runtime to 1.20 and 1.35, respectively. Therefore, we use 3 views in our main experiments as it captures most of the performance benefits of multi-view prediction while maintaining a favorable efficiency–accuracy trade-off.

*Table 10.* Ablation on camera embedding. We compare our proposed camera embedding against a baseline that flattens camera extrinsics and projects them with an MLP (*Flatten Cam*). Results are reported on RoboTwin and our real-world dataset. SSIM, AbsRel, RMSE, $\delta_1$, CD, and EMD are scaled by $10^2$.

| Method | Appearance | | | Depth | | | Point Cloud | |
|---|---|---|---|---|---|---|---|---|
| | PSNR↑ | SSIM↑ | FVD↓ | AbRel↓ | RMSE↓ | $\delta_1$ ↑ | CD↓ | EMD↓ |
| *RoboTwin* | | | | | | | | |
| Flatten Cam | 22.57 | 89.6 | **21.26** | 3.27 | 15.62 | 93.9 | 11.36 | 12.45 |
| Ours | **22.91** | **90.2** | 21.93 | **2.60** | **12.30** | **97.4** | **6.51** | **9.90** |
| *Real-world Dataset* | | | | | | | | |
| Flatten Cam | **22.19** | 89.5 | **21.36** | 21.43 | 25.65 | 80.60 | 17.75 | 16.82 |
| Ours | 21.82 | **89.98** | 23.08 | **20.79** | **25.11** | **82.18** | **13.06** | **14.37** |

*Table 11.* Effect of the number of views on RLBench average success rate and time cost.

| Num Views | 1 | 2 | 3 | 4 | 5 |
|---|---|---|---|---|---|
| RLBench (%) | 68.6 | 71.5 | 72.6 | 72.9 | 73.1 |
| Time cost | 0.78 | 0.85 | 1.00 | 1.20 | 1.35 |

## H. Failure Case

This section summarizes representative scenarios where our method underperforms. Overall, failures mainly arise from (i) ambiguous action direction under partial observations, and (ii) imprecise spatial grounding that leads to near-miss contacts in fine manipulation.

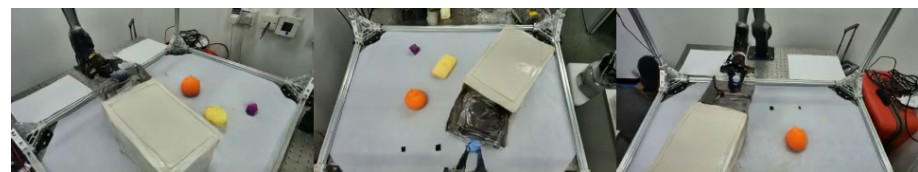

*Figure 9.* Failure case in the open-drawer task: our method successfully localizes the drawer handle but pulls in an incorrect direction, likely due to ambiguity in the optimal pulling direction under occlusion and limited viewpoints.

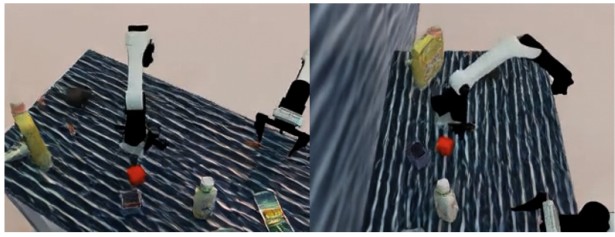

*Figure 10.* Failure case on RoboTwin (contact-sensitive task): the policy roughly reaches the target area but the end-effector is spatially misaligned and misses the red target block.

In both cases, the generated future captures high-level intent (e.g., reaching the handle/target region), but small errors in geometry or contact timing can be amplified during execution. We leave improving spatial precision and contact robustness (e.g., via higher-resolution geometry, stronger contact-aware constraints, or closed-loop correction) as future work.

## I. Limitation

By introducing action guidance and enabling test-time action optimization, our approach strengthens consistency between actions and the generated multi-view videos, yet it has limitations. First, test-time optimization requires multiple steps,

increasing latency, and can reduce real-time responsiveness. Second, the method depends on accurate camera and robot calibration; changes in extrinsics or robot kinematics can reduce consistency and action quality.

## J. Additional Qualitative Results of 4D Generation

In this section, we provide additional qualitative results on the RLBench dataset in Fig. 13, Fig. 14, and Fig. 15. More visualizations and longer rollouts are included in the accompanying supplemental video.

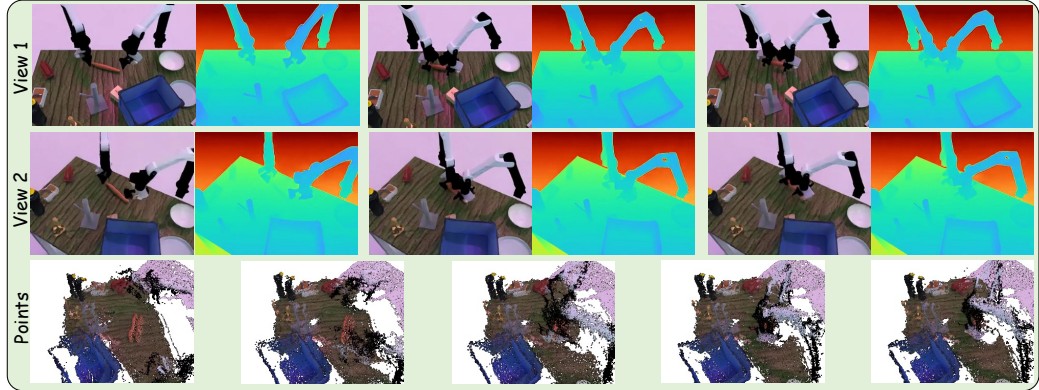

(a) 4DGen: double view generation on Robotwin

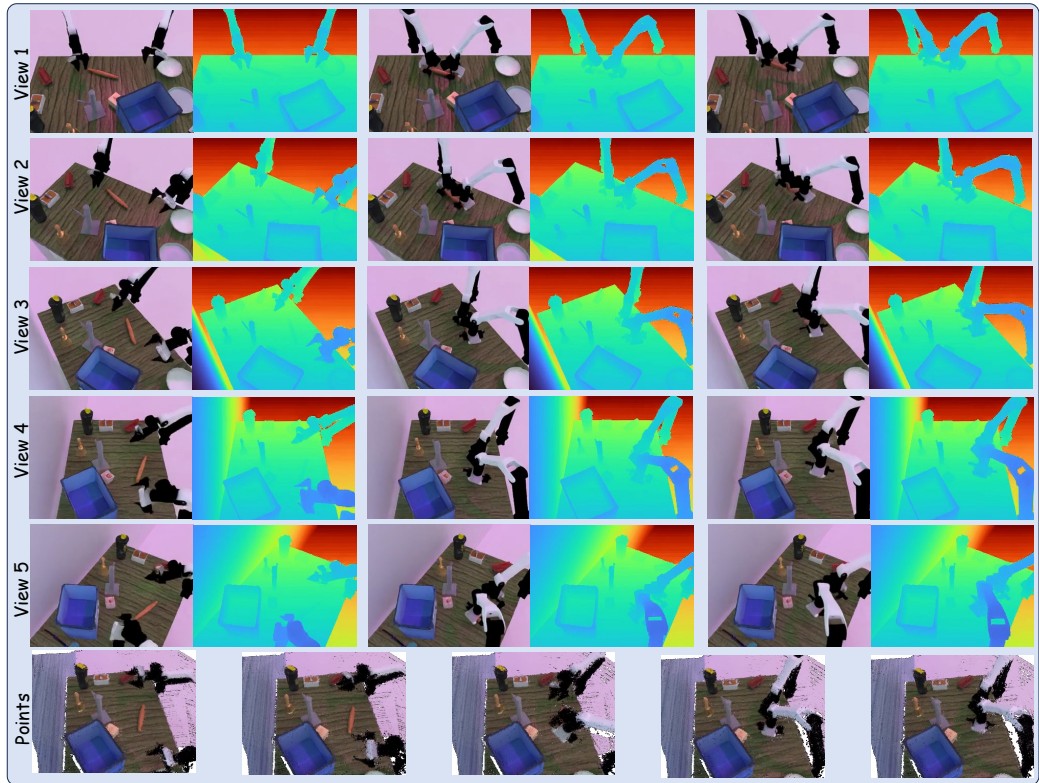

(b) Ours: five view generation on Robotwin

*Figure 11.* Multiview generation results by Mode-2 strategy on RoboTwin Dataset and corresponding results from 4DGen

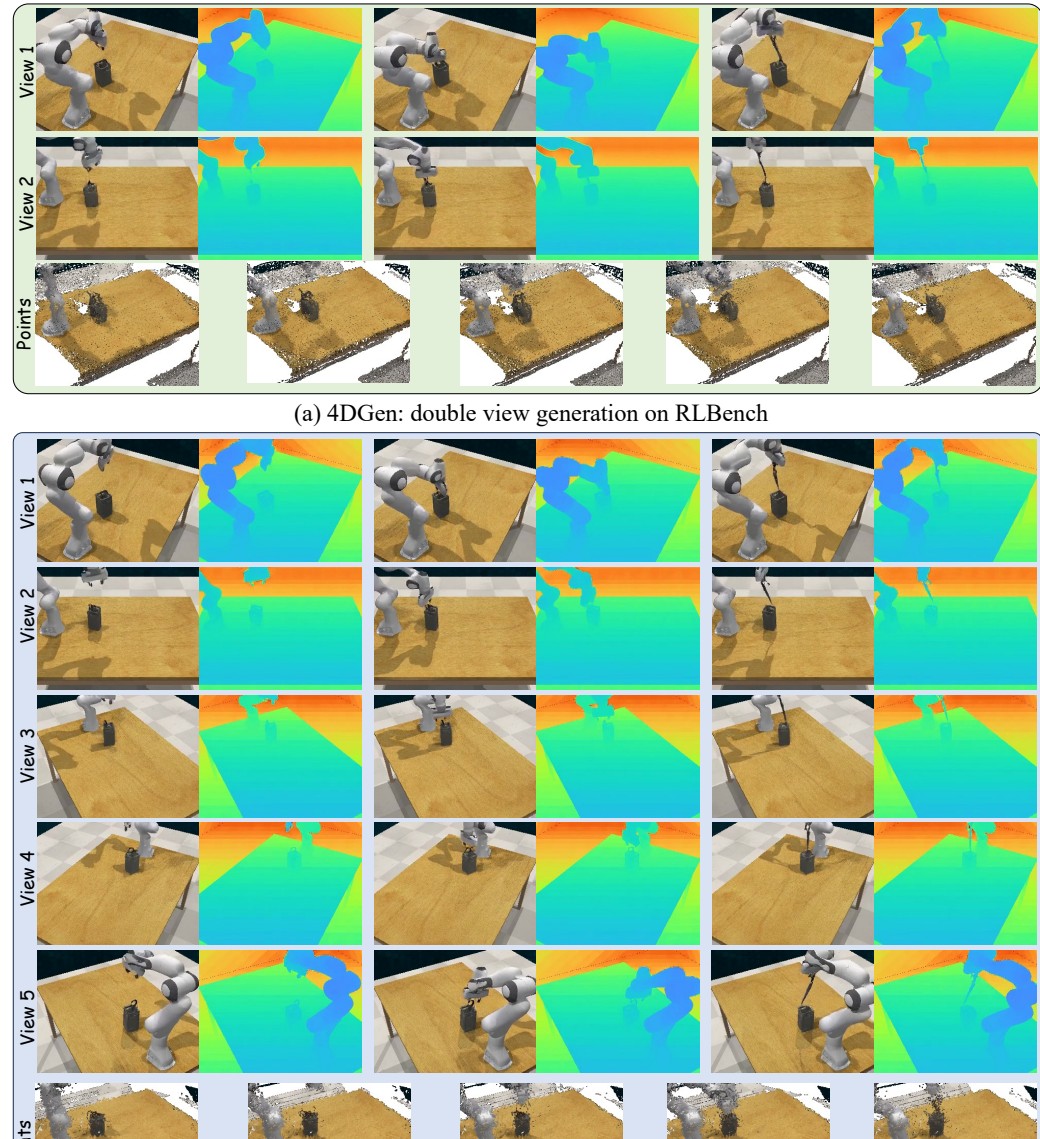

(a) 4DGen: double view generation on RLBench

(b) Ours: five view generation on RLBench

*Figure 12.* Multiview generation results by Mode-1 strategy on RLBench Dataset and corresponding results from 4DGen

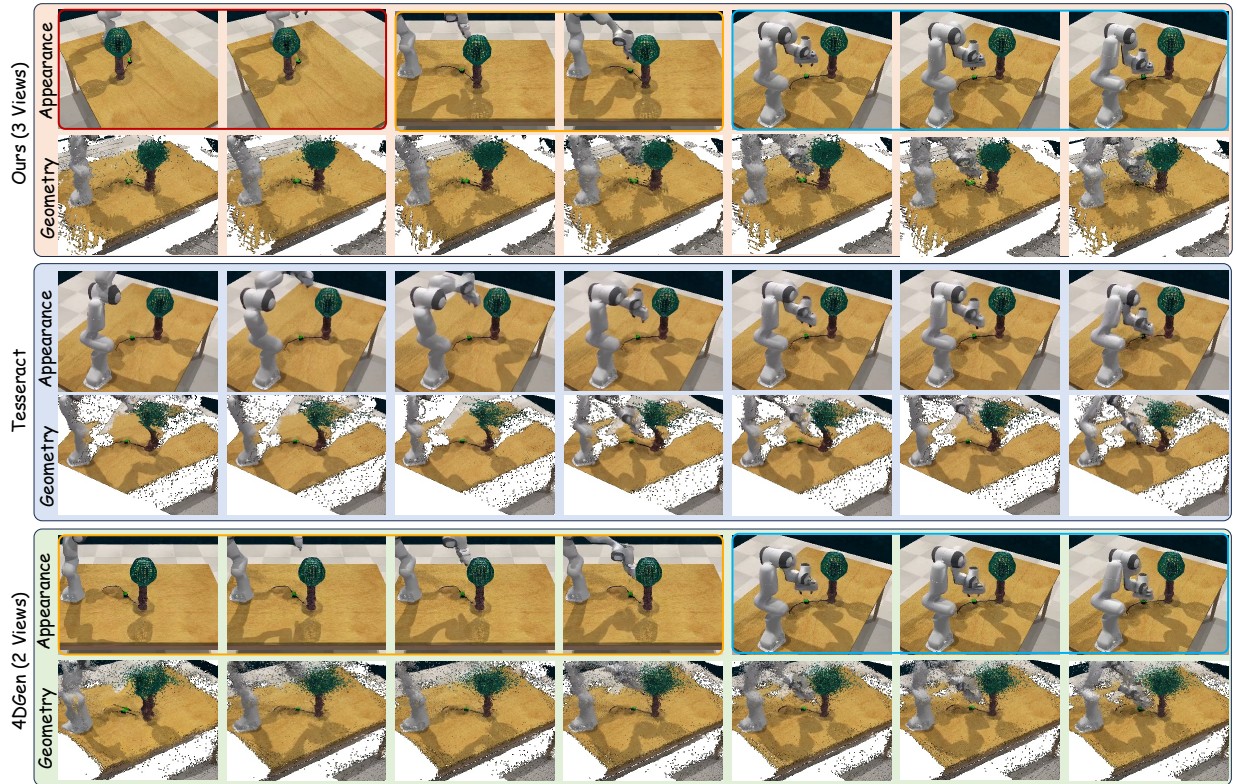

Figure 13. Qualitative Generation Results on RLBench Dataset.

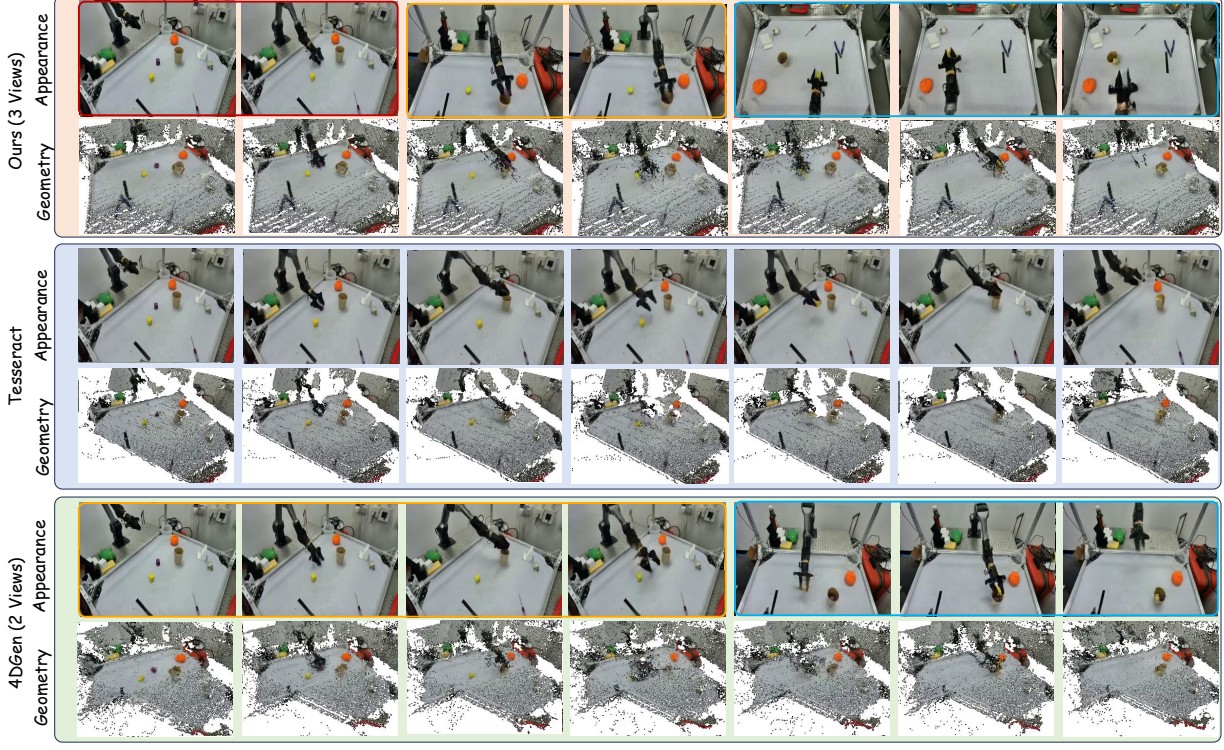

Figure 14. Qualitative Generation Results on Real-World `place cubes into cup` task.

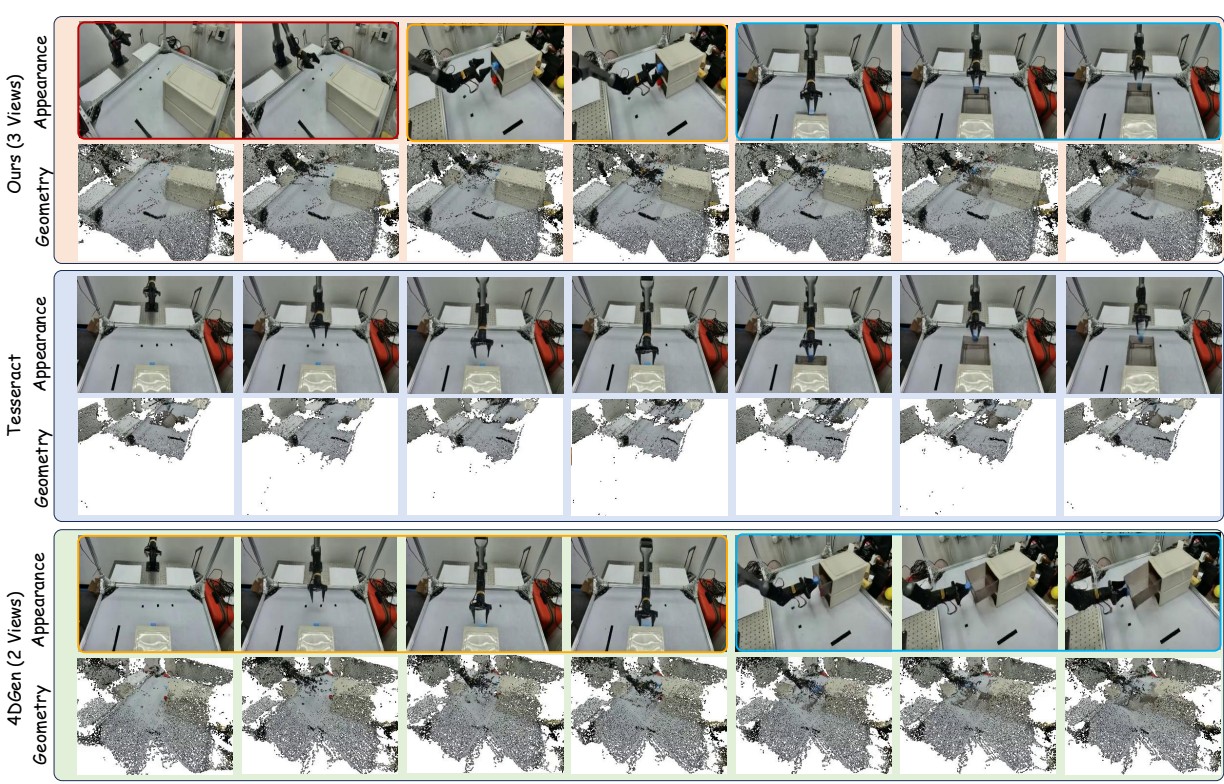

*Figure 15.* Qualitative Generation Results on Real-World `open drawer` task.

