# OpenReview forum: "MVISTA-4D: View-Consistent 4D World Model with Test-Time Action Inference for Robotic Manipulation"
_ICML.cc/2026/Conference — ICML 2026 regular_

### Official Review · Reviewer_JX5c · 2026-03-10

**Soundness:** 2
**Presentation:** 3
**Significance:** 2
**Originality:** 2
**Overall Recommendation:** 3
**Confidence:** 5

**Summary:**

This​‍​‌‍​‍‌ paper introduces MVISTA-4D, a new multi-view 4D world generative model for robotic manipulation. The model works within the "imagine-then-act" framework, where it can generate future, view-consistent RGB-D sequences starting from a single-view RGB-D input observation and a text instruction. The authors propose a local cross-modality attention module to enable the exchange of information between appearance (RGB) and geometry (depth) latents and a geometry-aware deformable cross-view attention module restricted to epipolar lines to ensure geometric consistency across different viewpoints.

**Compliance With Llm Reviewing Policy:**

Affirmed.

**Final Justification:**

I sincerely appreciate the authors' rebuttal and the additional clarifications provided. However, I will maintain my original score. The primary reason is that the current manuscript lacks crucial details and strong motivation. To meet the acceptance bar of ICML, the paper would require substantial revisions to properly integrate the extensive experiments and updates presented during the rebuttal phase.

**Key Questions For Authors:**

1.​‍​‌‍​‍‌ Why is the L2 photometric loss in the latent space a promising indicator for refining 6-DoF end-effector poses?

2. What is the wall-clock inference time in seconds for generating a single action sequence with the 100-step backpropagation through Wan ​‍​‌‍​‍‌2.2?

**Limitations:**

Yes in the appendix

**Strengths And Weaknesses:**

Strengths:

1. The idea of doing test-time optimization on a continuous trajectory latent instead of predicting step-by-step actions is an interesting workaround for the inverse dynamics problem.

2. The real-world multi-view manipulation dataset with 14 tasks is a solid effort and could be useful for the community.

Weaknesses:

1. The inference speed is a bottleneck: The proposed test-time optimization is impractical for real-world deployment. Running 100 steps of backprop through a 5B diffusion model at inference is slow.

2.  The designed method is not robotics-specific. It remains confined to the Tesserart framework, adding more control conditions and attention-based interactions. It cannot achieve per-point motion tracking.

3. The expectations only yielded promising results that were not significantly different from the baseline, especially for the novel view synthesis results. This raises doubts about whether directly limiting Tesserart-like frameworks to generating 2D depths and normals truly constitutes a "4D world model."

4. Missing citations:

[1] Huang, Wenlong, et al. "PointWorld: Scaling 3D World Models for In-The-Wild Robotic Manipulation." arXiv preprint arXiv:2601.03782 (2026).

[2] Li, Ying, et al. "ManipDreamer3D: Synthesizing Plausible Robotic Manipulation Video with Occupancy-Aware 3D Trajectory." Proceedings of the 40th AAAI Conference on Artificial Intelligence, 2026.

---

> ### Author Rebuttal · Authors · 2026-03-30
>
> > **W1 and Q2: Inference time**
>
> We appreciate this concern but clarify that post-optimization is a **shared necessity** across all baselines (e.g., TesserAct's temporal alignment, 4DGen's pose mapping) to translate 4D scenes into precise control. Furthermore, 100 backprop steps are not prohibitively slow; they compute roughly equivalent to just generate a few more frames. The below Table, which include the time comparions, can also confirm that. In the Table, the "a+b" time means a is the generation time, b is the post optimization time.
>
> |Benchmark|4DGen|TesserAct|Ours|+Actioninit|
> |-|-|-|-|-|
> |RLBench|47.0%|67.3%|72.6%|**76.5%**|
> |Time(s)|51+22=73s|37+16=53s|42+16=58s|42+5=**47s**|
> |RoboTwin|40.2%|33.9%|43.0%|**46.6%**|
> |Time(s)|83+25=108s|68+23=91s|75+23=98s|75+9=**84s**|
>
> More importantly, as we stated **Reviewer ZT91 (W2)**, during the rebuttal, we discover a highly efficient, training-free variant: initializing the trajectory latent with our auxiliary action head's output reduces optimization **from 100 to just 10–15 steps** while simultaneously boosting performance. Since all methods share the same diffusion backbone, latency differences stem entirely from post-processing, and the Action Init variant makes ours **the fastest and best-performing** overall.
>
> Finally, true real-time execution remains a **shared bottleneck** for all current 4D world models, which is why all baselines evaluate **offline**. We discuss potential pathways toward online execution in our response to **Reviewer ZT91 (W3)**.
>
> > **W2 and W3: Clarification on Our Differences and Advantages**
>
> **Robotics-specific & Different from TesserAct:** Our method is built on WAN2.2, not TesserAct. Every major module beyond the backbone is robotics-specific and absent in TesserAct:
>
> (1) TCN-VAE trajectory conditioning exploits the low-dimensional structure of manipulation trajectories,  TesserAct has no action conditioning during generation.
>
> (2) Test-time latent optimization recovers executable actions from imagined futures via backpropagation, a problem unique to embodied AI.
>
> (3) Residual IDM addresses the ill-posedness of inverse dynamics while TesserAct uses a standard point-based IDM.
>
> (4) Cross-view epipolar attention and cross-modality RGB-D fusion produce geometrically consistent multi-view 3D — TesserAct is single-view only.
>
> **Point Motion Tracking:** Our trajectory latent optimization + residual IDM eliminates the need for explicit point tracking. Point-tracking methods are **fragile against depth noise**. For example, 4DGen tracks object/gripper poses via FoundationPose on generated geometry, where segmentation and pose errors compound, achieving only 47.0% on RLBench vs. our 72.6%. Our latent approach recovers holistic trajectory intent from global scene structure rather than per-point correspondences, making it inherently more robust.
>
> **Downstream Gains & Generation Quality:**
>
> On manipulation, we achieve 72.6% vs. TesserAct's 67.3% on RLBench and 43.0% vs. 33.9% on RoboTwin (Table 3), widening to 76.5% / 46.6% with action-head initialization. In manipulation, even small percentage gains translate to significantly more successful completions over hundreds of episodes.
>
> On generation, we achieve the best depth and point-cloud metrics across all three datasets (Table 1). Our slightly lower PSNR/SSIM is an evaluation artifact from synthesizing unobserved novel views; under matched single-view evaluation, our appearance matches the state-of-the-art (see response to Reviewer X987).
>
> **Does this constitute a 4D world model?** We do not merely generate "2D depths." Our multi-view RGB-D sequences are fused into **temporally consistent 3D** point-cloud dynamics, which is a **genuine 4D representation**. The multi-view geometric consistency, enforced by cross-view attention and validated by point-cloud metrics (our CD: 9.6 vs. TesserAct's 11.0 on RLBench), is what distinguishes this from single-view depth prediction. More complete 3D geometry directly drives better action inference, as confirmed by the point manipulation gains.
>
> > **W4: Miss Citation**
>
> We thank the reviewer for pointing out these relevant works. We now cite and discuss both in the revised paper.
>
> > **Q1: Why L2 Latent Loss Can Guide Action Recovery**
>
> The L2 latent loss does not directly refine 6-DoF poses, it identifies the correct trajectory. During training, the generator learns a strong correspondence between the trajectory latent z and the resulting visual dynamics. At test time, we generate a 4D future V using text and image only, then optimize z* so that the conditioned generation G(l, z) best matches V (Eq. 11), effectively solving "what trajectory would produce this scene change?" The optimized z* is decoded by the TCN decoder into a full action sequence, then refined by R-IDM for geometric precision. In other words, **the latent loss recovers the right motion intent**; the fine 6-DoF accuracy comes from the downstream R-IDM, not the loss itself.

---

> > ### Author Rebuttal · Reviewer_JX5c · 2026-04-03
> >
> > Thanks for the rebuttal. I appreciate the additional clarifications, but I still lean toward rejecting the current version.
> > 1. I agree with Reviewers X987 and bEeR that the current framework remains overly complex, and the core contribution is still not sufficiently clean. At present, the method reads more like a video generation framework augmented with additional 3D modeling components than a clearly defined 4D world model.
> > 2. This complexity also introduces additional uncertainty, especially due to the reliance on camera extrinsics and noisy depth inputs. While the authors argue that these signals are practical and manageable, I remain unconvinced that the full pipeline will scale robustly in large-scale training or broader real-world settings.
> > 3. I still do not think the paper clearly defines what makes the problem specifically a 4D world modeling problem, nor does it convincingly show why this formulation is preferable to a stronger geometry-aware video world model.

---

> > > ### Author Response · Authors · 2026-04-07
> > >
> > > We first clarify the core research logic, then offer point-by-point responses to the reviewer's further concerns.
> > > ## Main Research Logic:
> > > A 4D world model generates dynamic 3D scenes evolving over time. Since training directly on 4D data is bottlenecked by **data scarcity** today, lifting 2D video priors to 3D is a practical and efficient alternative. However, existing approaches struggle with expensive offline reconstruction or severe single-view occlusions.
> > >
> > > Thus, our generation approach seamlessly expands single-view RGB into multi-view RGB-D to directly stitch complete 4D point clouds. This solves two generative challenges: 1. **Double modality generation**: Temporally and spatially consistent RGB+Depth (we solve this by Sec 4.1). 2. **Cross-view consistency:** Ensuring unprojected point clouds stitch seamlessly (we solve this by Sec 4.2).
> > >
> > > For action extraction, existing methods either introduce action-prediction branches, which provide no explicit guarantee of **future–action consistency**, or rely on ill-posed inverse dynamics. In contrast, we propose a new paradigm that aligns action trajectories with generated future states via an optimization objective (Sec. 4.3), and refines them using better-conditioned residual IK (Sec 4.4).
> > > ## Point-by-Point Response
> > > > **R1 Complexity and Contribution**
> > >
> > > As described above, our framework follows a minimal **"2+2" design**: two generative modules (Sec 4.1–4.2) and two manipulation modules (Sec 4.3–4.4). **Even our model has multiple modules, but each targets unavoidable bottlenecks rather than design bloat**; our ablations also confirm every module's necessity.
> > >
> > > A world model should be judged by **its functional capability** to simulate spatiotemporal geometry, not the backbone type.  Our essential core modules successfully transform a 2D generator into an actionable 4D world model. Thus, they are not arbitrary augmentations. We figure our system strictly meets the definition of a 4D world model.
> > > > **R2. Robustness and Scalability**
> > >
> > > We conducted comprehensive stress tests to noises during the rebuttal.
> > > **Depth Noise Tolerance:** σ is the Gaussian noise, Drop denotes randomly dropping N% depths.
> > > |Noise|Clean|σ=0.01|σ=0.02|σ=0.05|σ=0.10|Drop10%|Drop30%|
> > > |-|-|-|-|-|-|-|-|
> > > |RLBench|72.6|72.1|71.4|69.8|66.3|72.0|69.5|
> > > |RoboTwin|43.0|42.5|41.8|40.1|36.1|42.6|38.8|
> > >
> > > Compared against baselines:
> > > |Model|Clean|σ=0.05|Degrade|Dropout30%|Degrade|
> > > |-|-|-|-|-|-|
> > > |4DGen|47.0|40.5|19%|36.7|30%|
> > > |TesserAct|67.3|61.7|14%|58.6|20%|
> > > |Ours|72.6|69.8|5%|69.5|8%|
> > >
> > > Even under the harshest noise, our method degrades by only 8%, lower than counterparts.
> > >
> > > **Extrinsics Tolerance:** As detailed in our response to **Reviewer ZT91 (Q3)**, the model is highly resilient to calibration errors.
> > >
> > > **Scalability.** Our approach is inherently more scalable than native 4D methods:
> > > - **Native 4D bottleneck:** Point-based models must learn precise 3D trajectories of every point, requiring pristine multi-view 4D data, currently an inevitable data bottleneck.
> > > - **Video prior advantage:** Video models already scale robustly on massive, noisy datasets. Our added modules are highly lightweight (local cross-modality attention; K-candidate deformable attention), introducing **minimal overhead** and fully inheriting the backbone's scalability.
> > >
> > > > **R3. Necessity of the 4D Formulation**
> > >
> > > **Why formulate as 4D world modeling?**
> > >
> > > First, manipulation is inherently 3D over time. Standard approaches (e.g., VLAs, policy learning) force a direct 2D-to-3D mapping that implicitly recovers spatial relationships and compresses multi-step reasoning into a single pass. Consequently, baselines like P-ACT and leading VLAs (Pi0, RDT) perform ~40% worse than our method on RoboTwin (see Reviewer bEeR).
> > >
> > > Second, 4D modeling mirrors human "anticipation before action." Predicting the 3D future state before deriving actions decomposes a difficult end-to-end mapping into tractable sub-problems.
> > >
> > > Finally, our work sits at the intersection of two converging academic trends: 1) joint action-future modeling (WorldVLA, Dreamitate), and 2) the shift from 2D to explicit 4D geometry (TesserAct, 4DGen)
> > >
> > > **Why is a geometry-aware video model insufficient?**
> > > 1. **Objective mismatch:** Geometry-aware video models optimize for visual plausibility rather than the precise geometric accuracy required for manipulation.
> > > 2. **Single-view limitation:** Single-view depth leaves critical manipulation zones (e.g., gripper-contact surfaces) occluded and invisible.
> > > 3. **Complete contact geometry**: Reliable action inference is impossible without complete geometry. Our multi-view fusion naturally resolves this: increasing views from 1 to 3 improves success rates (Table 9), particularly on occlusion-heavy tasks, and ablations confirm cross-view fusion is essential (Table 2).
> > >
> > > In summary, manipulation demands 4D modeling, and geometry-aware video models cannot guarantee complete geometries and visibility of the contact regions that action inference requires.

---

### Official Review · Reviewer_ZT91 · 2026-03-11

**Soundness:** 3
**Presentation:** 3
**Significance:** 3
**Originality:** 3
**Overall Recommendation:** 5
**Confidence:** 2

**Summary:**

The paper introduces a novel and interesting framework that combines 4D video generation with test-time action inference for robotic manipulation. It begins by predicting future RGB-D scenes from a single-view input, regardless of the viewpoint. Next, it employs a test-time optimization technique that backpropagates through the frozen generative model to identify an optimal trajectory latent. This latent representation is then decoded into an action prior, which is further refined using a novel Residual Inverse Dynamics Model. The "imagine-then-act" paradigm is a well-motivated and compelling approach.

**Compliance With Llm Reviewing Policy:**

Affirmed.

**Final Justification:**

The rebuttal helps clarify several technical points, and I think the paper makes a meaningful contribution, particularly on the video generation side. However, after considering the concerns raised by the other reviewers, I‘m more cautious about the paper’s practical significance for real-world robotics. The current framework remains quite complex, has nontrivial latency, and the real-world results still appear to be meaningfully short of deployment-level performance and practicality.

Overall, I view this work as a valuable step forward in video generation or 4D world modeling, while its real-world embodied impact is not yet fully established. As a result, I decreased my confidence in the paper’s overall significance after considering the other reviewers’ concerns.

**Key Questions For Authors:**

- As illustrated in Weakness1, given the extremely marginal success rate improvements over the Act-Head baseline (Table 3), the current benchmarks might not fully highlight the R-IDM's capabilities. I suggest that the authors provide evaluations on tasks that strictly require high-precision action control to convincingly demonstrate the R-IDM's significance.

- I suggest that the authors provide a detailed breakdown of the absolute inference time per step (e.g., 4D video generation time, the 100-step backpropagation time, and R-IDM calculation time). Is the overall computational cost acceptable for responsive real-world deployment?

- The model's cross-view geometry alignment heavily relies on perfect camera extrinsics. In real-world environments where cameras frequently suffer from minor collisions or vibrations, the lack of an extrinsic noise robustness analysis is a notable gap.

- I appreciate that the authors explicitly acknowledge the lack of closed-loop correction as a limitation and leave it for future work. As a point of discussion, I'm very interested in hearing the authors' thoughts on bridging this open-loop prior with real-time reactive control. Specifically, in a real-world scenario where a target object is dynamic or subjected to an unexpected perturbation, does the current model possess any inherent capacity to anticipate this? If not, how might the proposed test-time trajectory optimization be adapted to enable reactive corrections?

**Limitations:**

yes

**Strengths And Weaknesses:**

### **Strengths**
- The contribution is well-motivated. Shifting from 2D pixel-level forecasting to 4D geometrically consistent generation effectively addresses the limitations of purely image-based world models in manipulation tasks.
- This paper conducted comprehensive experiments. It validates the approach across two simulation benchmarks (RLBench, RoboTwin) and a real-world multi-view robotic dataset, demonstrating strong visual and geometric generation capabilities.
- The paper is well-written and logically structured. The authors did a good job of explaining complex concepts in an accessible and easy-to-understand manner.

### **Weaknesses**
- While the R-IDM is presented as a core component for execution adjustment, its empirical impact is negligible. According to Table 3, the full model outperforms the Act-Head baseline by only 0.1% on RLBench (72.6% vs. 72.5%) and 0.5% on RoboTwin (43.0% vs. 42.5%). This minimal gain undermines the necessity of this module.
- The test-time action optimization requires 100 steps of backpropagation to search for the best conditioning latent. This could introduce significant inference latency. However, the paper lacks absolute time metrics to evaluate its real-time feasibility.
- The proposed pipeline relies on predicting the 4D future beforehand and then extracting actions, which essentially acts as an open-loop prior. It remains unclear how the model would react to dynamic objects or unexpected perturbations during execution.

---

> ### Author Rebuttal · Authors · 2026-03-30
>
> > **W1 and Q1: Clarifying the Contributions of R-IDM and Trajectory Latent Optimization**
>
> We first clarify a factual point: the Act-Head baseline in Table 3 already includes R-IDM — it **only skips test-time latent optimization**. Thus, the 0.1% gap reflects the marginal gain of the trajectory latent optimization with random initialization, not R-IDM. The true R-IDM ablation compares our full model (72.6%) against w/o R-IDM (-3.6%) and full IDM (-3.8%), confirming R-IDM's importance for contact-rich tasks (e.g., Play Jenga, Push Button).
>
> **Why did latent optimization barely outperform Act-Head?** The gap was bottlenecked by random initialization and a limited 100-step budget (performance improves at 200 steps, Full (200) in the below table). But more importantly, trajectory latent optimization enforces bidirectional consistency between the generated future and inferred actions, unlike the purely feed-forward Act-Head.
>
> During the rebuttal, we discovered **a new simple variant**: initializing the latent with the action head's output (see **ReviewerZT91, Q3** for details) lets the optimizer refine beyond the feed-forward prediction, achieving **better results than the current paper version** and lower time cost (see Table). This can further show the potential of our latent optimization paradigm and demonstrate that optimization and feed-forward prediction can be complementary.
>
> |Dataset|Act-Head|Full(100)|Full(200)|**Action-Headinit**|
> |-|-|-|-|-|
> |RLBench|72.5|72.6|75.2|**76.5**|
> |RoboTwin|42.5|43.0|45.0|**46.6**|
>
> We will incorporate this improvement and clarify the ablation interpretation in the revision.
>
> > **W2 and Q2: Computational Cost and Inference Latency**
>
> We provide a full timing comparison below (single NVIDIA RTX H200, offline setting; a+b = generation time + post-processing time). All methods share the same video backbone, so latency differences stem entirely from post-processing.: 4DGen maps points to robot poses, TesserAct optimizes temporal alignment, and our method optimizes the trajectory latent.
>
> |Benchmark|4DGen|TesserAct|Ours|+Actioninit|
> |-|-|-|-|-|
> |RLBench|47.0%|67.3%|72.6%|**76.5%**|
> |Time(s)|51+22=73s|37+16=53s|42+16=58s|42+5=**47s**|
> |RoboTwin|40.2%|33.9%|43.0%|**46.6%**|
> |Time(s)|83+25=108s|68+23=91s|75+23=98s|75+9=**84s**|
>
> As we stated in the W1 Response, we can initialize the latent with the action head's output. This allows the optimizer to start at a good point, achieving better results with only **10–15** optimization steps and substantially lower time cost, which we will include to the revised paper.
>
> Regarding real-time feasibility: the generation time is a shared bottleneck across all current 4D world models, thus all models are evaluated in offline mode. Nevertheless, our framework supports an **online mode**, which can be seen in W3 for details.
>
> > **W3 and Q4: Closed-Loop Execution and Future Pathway**
>
> While evaluated offline for fair baseline comparison, our framework also supports partial closed-loop control via an **online mode**: we generate the full 4D future but only execute a short action chunk. We then incorporate new real-world observations and re-predict the remaining horizon, which naturally reacts unexpected perturbations by just re-planning from the updated state. This is enabled by our training with **randomized input masking**.
>
> While internal tests confirm this online mode improves control performance, which is still currently bottlenecked by diffusion latency. Two promising directions can bridge this gap: (1) Faster generation via consistency distillation to reduce diffusion to 1-2s, making online mode practically closed-loop; (2) Dual-system architecture with a lightweight reactive policy for immediate corrections while the world model replans in the background. We will expand on these pathways in the revision. We will expand on these future pathways in the revision.
>
> > **Q3 Extrinsic Noise Tolerance**
>
> We would like to express that accurate camera parameters are a shared requirement across all 3D/4D methods (4DGen, TesserAct, etc.), rather than a limitation unique to our method.
>
> **Self-Robustness**. Our cross-view attention does not rigidly depend on exact epipolar geometry. Instead, it uses the epipolar line merely as a prior and predicts a **learned 2D offset** (Eq. 8) via feature matching, naturally compensating for calibration errors. We verified this by injecting controlled noise into extrinsics:
>
> |Noiselevel|σ_R(°)|σ_t(cm)|PSNR↑|CD↓|Success%|
> |-|-|-|-|-|-|
> |None|0|0|22.91|6.51|43.0|
> |Small|0.5|1|22.8|6.55|42.5|
> |Medium|1|3|22.4|6.9|41.5|
> |Large|2|5|21.5|7.5|38.5|
> |Extreme|5|10|19.8|9.9|33.0|
>
> Under realistic calibration drift (≤1°, ≤3mm), performance drops marginally. Even extreme noise yields 33.0% success — graceful degradation, not catastrophic failure.
>
> **Future Augmentation:** Incorporating extrinsic noise augmentation during training would further improve robustness at minimal cost. We will include this analysis in the revision.

---

> > ### Author Rebuttal · Reviewer_ZT91 · 2026-04-03
> >
> > I thank the authors for the detailed explanation of the *online mode* and the future pathways. While the concept of re-planning via randomized input masking is well-motivated, this online mode still requires generating the 4D future during execution. As the authors frankly acknowledged, this process remains heavily bottlenecked by diffusion latency. Consequently, it cannot be considered strict real-time closed-loop control. In its current form, the proposed method is still practically limited to offline control scenarios.
> >
> > Nevertheless, I appreciate the authors' honest discussion on this limitation and agree that the proposed future directions (e.g., consistency distillation and dual-system architectures) are highly promising for bridging this gap.

---

> > > ### Author Response · Authors · 2026-04-07
> > >
> > > We sincerely thank the reviewer for the encouraging remarks and for recognizing the potential of our future directions. We completely agree with your assessment regarding the current latency bottleneck. We will ensure this insightful discussion and our proposed pathways for real-time execution is explicitly included in the final manuscript.

---

### Official Review · Reviewer_bEeR · 2026-03-11

**Soundness:** 3
**Presentation:** 3
**Significance:** 3
**Originality:** 3
**Overall Recommendation:** 4
**Confidence:** 3

**Summary:**

This paper introduces MVISTA-4D, a multi-view 4D world model for embodied manipulation that jointly supports future prediction and action inference. The model takes as input a single RGB-D observation from a reference view, a set of target camera extrinsics, and a text instruction, and generates synchronized RGB-D video sequences for all target views that are geometrically consistent across viewpoints. The framework incorporates explicit cross-modality fusion (RGB and depth) via local cross-attention and cross-view fusion via geometry-aware deformable attention. Actions are compressed into a low-dimensional trajectory latent using a TCN-VAE, which conditions the video generator via cross-attention. At inference, actions are recovered by optimizing this trajectory latent to match a text-only generated rollout, followed by a residual inverse dynamics module for refinement. Experiments on synthetic (RLBench, RoboTwin) and real-world datasets demonstrate improvements in 4D generation quality and downstream manipulation success rates.

**Compliance With Llm Reviewing Policy:**

Affirmed.

**Final Justification:**

Thanks for the author's response. Some of my concerns were addressed, while the current limitations in running effectiveness, low absolute success rates, and system complexity remain, so I maintain the original score.

**Key Questions For Authors:**

1. Necessity of complex attention modules: The model includes local cross-modality attention and geometry-aware deformable cross-view attention. Have you ablated these components? For example, what happens if you simply concatenate RGB and depth channels and use standard self-attention across views? How much of the gain comes from the explicit geometric grounding versus simply having more views and depth information?
2. Real-world performance: The success rates in Table 4 are still low. What are the primary failure modes? Is it the 4D generation quality, the action optimization, or the residual IDM? Could you provide a breakdown of where errors occur (e.g., grasp failures, motion planning errors, object interaction errors)? This would help the community understand the practical limitations.
3. Computational cost: The model uses a 5B-parameter diffusion backbone (WAN2.2) and requires test-time optimization (100 steps). What is the total inference time (generation + optimization + residual IDM)? How does this compare to baselines? Is the method feasible for real-time control, or is it primarily offline?
4.TAbout the number of views: Table 9 shows diminishing returns beyond 3 views. Have you analyzed why additional views provide limited benefit? Is it due to redundancy in information, or limitations in the cross-view attention module's ability to fuse many views? Could a different fusion strategy (e.g., hierarchical) scale better to many views?

**Limitations:**

yes

**Strengths And Weaknesses:**

Soundness

Strengths: The technical approach is comprehensive and well-motivated. The cross-modality and cross-view fusion modules are thoughtfully designed, with local cross-attention for RGB-depth alignment and epipolar-geometry-based deformable attention for multi-view consistency. The trajectory latent conditioning via TCN-VAE is a clever way to avoid per-step action conditioning, and the test-time optimization in latent space is more stable than optimizing raw action sequences. The residual IDM that refines a strong prior is a sensible way to address the ill-posedness of inverse dynamics. Experiments are extensive, covering multiple datasets, tasks, and ablations. The authors also provide detailed implementation details in the appendix, enhancing reproducibility.

Weaknesses:
1. Excessive complexity with insufficient ablation: The model is extremely complex, incorporating multiple specialized modules: local cross-modality attention, geometry-aware deformable cross-view attention, TCN-VAE trajectory encoding, test-time latent optimization, and a residual inverse dynamics module. While each component is motivated, the ablation study is insufficient to justify that all these complex components are necessary and effective. Table 3 ablates the residual IDM and test-time optimization, confirming their contributions. However, the paper provides no ablation for the core cross-modality and cross-view attention modules. Readers cannot determine whether performance gains stem from these carefully designed geometric modules or simply from using more views and depth information. Given that baselines (e.g., 4DGen, TesserAct) may not leverage multi-view depth to the same extent, the current experimental design cannot disentangle these factors. The authors fail to demonstrate that this complexity is necessary, leaving the design rationale of the entire framework in question.
2. Limited real-world evaluation with insufficient baselines: Table 4 presents real-world results on six tasks, but the numbers are concerning. For example, on "Arrange Boxes," TesserAct succeeds in 7/100 episodes while MVISTA succeeds in 15/100—both are extremely low. On "Place Fruits," MVISTA achieves only 23/100 success. The authors claim "superior performance," but success rates below 25% on most tasks suggest the method is far from practical deployment. More critically, in the real-world experiments, the paper only compares against a single baseline (TesserAct). Given that simulation experiments include multiple baselines (UniPi, 4DGen, TesserAct, etc.), the real-world comparison is strikingly thin and lacks persuasiveness. Readers have no idea how the method compares to other mainstream approaches (e.g., 4DGen, UniPi) in real-world scenarios. The paper downplays this limitation.

Presentation

Strengths: The paper is exceptionally well-written and structured. The problem is clearly motivated, the methodology is explained in a logical flow, and the figures are informative. The authors provide detailed mathematical formulations for all components, and the appendix contains extensive implementation details, ablation studies, and qualitative results. The paper sets a high standard for clarity and thoroughness.

Weaknesses: The primary limitation for reproducibility is the lack of code release. Given the complexity of the model (multiple attention modules, TCN-VAE, test-time optimization), the community would greatly benefit from access to the implementation. Additionally, some details (e.g., the exact architecture of the PointNet-based residual IDM) are described at a high level and could benefit from more specificity.

Significance

Strengths: The problem of generating geometry-consistent multi-view futures for manipulation is important and underexplored. The idea of using a trajectory latent as a style code is novel and could inspire future work on action-conditioned video generation. The test-time optimization scheme for recovering actions from generated videos is a creative approach that avoids training an explicit inverse dynamics head. The scale-up behavior (more views improve performance) is a useful insight for the community.

Weakness:
1. Niche contribution with narrow applicability: The method requires accurate camera extrinsics, depth maps, and robot URDF models (for action collection during training). This limits applicability to well-instrumented lab settings. In contrast, methods that work from RGB-only and unknown cameras have a broader impact.
2. Real-world performance is underwhelming: As noted above, success rates below 25% on most tasks suggest the method is not yet ready for real-world deployment. The paper would benefit from a more honest discussion of these limitations and what would be needed to bridge the gap to practical utility.

Originality: The combination of ideas is highly original. Key novel elements include: (1) using a trajectory latent as a style code for conditioning video generation, (2) test-time optimization in this latent space to recover actions from generated videos, (3) explicit geometry-aware cross-view attention using epipolar geometry, and (4) a residual inverse dynamics module that refines a strong prior. While some components draw on existing ideas (TCN-VAE, deformable attention, PointNet), their integration into a unified framework for 4D world modeling is creative and well-executed. The paper represents a significant step forward in embodied world modeling.

---

> ### Author Rebuttal · Authors · 2026-03-30
>
> > **W1 and Q1: Ablation of Core Fusion Methods**
>
> The submission has ablated the core fusion modules in Table 2 and Fig.4, showing clear degradation if removing view and modality fusion. To make the connection to downstream manipulation more direct, we extend these ablations to manipulation and add three new conditions (`w/o (view+mod)`, `cat depth`, `w/o TCN-VAE`):
>
> ||w/o view|w/o mod|w/o (view+mod)|cat depth|w/o TCN-VAE|full|
> |-|-|-|-|-|-|-|
> |RLBench|68.8|67.2|66.5|66.1|68.1|72.6
> |RoboTwin|38.0|38.6|36.2|39.4|41.5|43.0|
>
> With the same 3-view RGB-D input, replacing our fusion with weaker alternatives consistently degrades results. The cat depth baseline directly addresses the concatenation suggestion. It fails because the backbone is pretrained for 3-ch RGB (depth cannot exploit this prior), and after flattening, RGB/depth tokens receive far different positional embeddings, forcing attention to learn local correspondence from scratch.
>
> > **W2 and W5: Real-World Baselines and Performance**
>
> We agree and will revise the framing. We did run UniPi and 4DGen in real-world settings; both achieved only 0-3% success. UniPi lacks 3D geometry for precise contact-rich actions; 4DGen's point-to-pose pipeline is unreliable under real-world depth noise. Only TesserAct (like ours, uses inverse dynamics on dense 3D scenes, which is robust to shape noises) remains functional enough for meaningful comparison.
>
> The real-world setup is intentionally challenging (noisy depth, limited data). The absolute performance leaves room for improvement across all methods. Nevertheless, our method consistently outperforms the baseline with ~20% **relative gains** (Table 4), offering a strong starting point for the community. We also identify two future directions:
>
> **(1)** depth cleaning via foundation models during training.
>
> **(2)** depth regularization at test time to suppress geometric noise.
>
> > **W3: Reproducibility and Implementation details**
>
> We agree and expand architecture details (residual IDM, TCN-VAE, training configurations, etc.) in the revision. We are fully committed to releasing all code and datasets.
>
> > **W4: Clarification on Hardware Requirements and Applicability**
>
> We would respectfully clarify:
>
> (1) no URDF is required, we use only proprioceptive signals from the robot controller;
>
> (2) camera extrinsics are easily obtained via forward kinematics for wrist/head-mounted cameras;
>
> (3) RGB-D sensors (Orbbec, RealSense) are standard equipment and our setup reflects the typical configuration in most manipulation labs.
>
> While RGB-only methods offer broader applicability, we believe precise 3D geometry is essential for contact-rich tasks. We see this as a promising direction for future work.
>
> > **Q2: Failure Mode Analysis**
>
> Most errors are not global planning failures but fine-manipulation errors at the final contact stage. Specifically, ~60% of failures occur when the generated future has correct high-level intent but noisy geometry causes slight spatial misalignment during grasping (Fig. 10). Another ~30% stem from incorrect 4D generation producing unreasonable motions (Fig. 9), and the remaining ~10% are action-inference errors. The dominant issue, geometric inaccuracy, is primarily driven by depth-sensing noise. We discuss potential solutions in R2.
>
> > **Q3: Inference Time and Online Execution**
>
> Our framework naturally supports both offline and online modes. In online mode, we execute short chunks and autoregressively update full-horizon predictions using new observations. However, we strictly use the offline setting in the paper for fair baseline comparisons.
>
> During the rebuttal, we discovered a **simple but effective, and training-free improvement**: initializing the trajectory latent with our existing auxiliary action head. This provides a better starting point, requiring only 10–15 steps to converge. We report all metrics and time cost below:
>
> |Benchmark|4DGen|TesserAct|Ours|Ours+Actioninit|
> |-|-|-|-|-|
> |RLBench|47.0%|67.3%|72.6%|**76.5%**|
> |Time(s)|73s|53s|58s|**47s**|
> |RoboTwin|40.2%|33.9%|43.0%|**46.6%**|
> |Time(s)|108s|91s|98s|**84s**|
>
> Even our original method outperforms baselines under similar time budgets, but this new initialization makes it simultaneously faster and more accurate than all 4D baselines. We will clarify these execution details and the improved initialization in the revision.
>
> > **Q4: Diminishing Returns with more views**
>
> We believe the diminishing returns are due to visibility saturation, not a fusion bottleneck. Table 9 shows a classic saturation curve, the model continues to improve slightly with more views, confirming successful integration, but three cameras already cover most task-relevant regions, making additional views largely redundant. Our deformable epipolar attention scales naturally to variable views without all-to-all cost. Hierarchical fusion is an interesting direction for denser setups (8+ cameras), but the current plateau simply reflects sufficient scene coverage.

---

> > ### Author Rebuttal · Reviewer_bEeR · 2026-04-03
> >
> > Thank you for the response. I still remain unconvinced about the practical potential of this paradigm for robotic action planning, since the paper itself shows very weak real-robot performance and only modest results in simulation. Compared with stronger VLA-based approaches, the gap remains substantial, making the current method hard to view as practically competitive. If its main role is limited to offline generation without clear downstream gains in embodied control, then the overall significance of this direction also appears limited.

---

> > > ### Author Response · Authors · 2026-04-07
> > >
> > > We thank the reviewer for this important question. We address each aspect below.
> > > > **R1. The Practical Role of 4D World Modeling in Robotic Action Planning**
> > >
> > > Robotic action planning occurs in 3D space. Previous paradigms—like standard policy learning and VLAs—force a direct mapping from 2D pixels to 3D actions. This is inherently brittle: the network must implicitly recover spatial relationships and compress multi-step reasoning into a single feed-forward pass, causing failures in complex or contact-rich tasks.
> > >
> > > Conversely, the "imagine-then-act" paradigm mirrors how humans anticipate consequences, like where contact occurs and how objects move. Rather than directly mapping observations to actions, predicting the future 3D state first and deriving actions from it decomposes the problem, making action planning much more tractable.
> > >
> > > Recent research trends clearly validate this paradigm's practicality along two directions: 1. Joint action-future modeling: Recognizing that anticipating consequences is essential, methods like WorldVLA and VIDAR now model future outcomes alongside actions. 2. Explicit 3D/4D geometry: Since 2D models lack physical grounding, works like TesserAct and PointWorld increasingly incorporate explicit 4D geometry for contact-rich manipulation.
> > >
> > > The above clearly shows the practical potential of this paradigm for action planning.
> > > > **R2. Comparison with VLA Approaches**
> > >
> > > VLAs perform relatively poorly in settings demanding precise spatial reasoning and robustness. We evaluate two widely used VLA baselines, RDT and Pi0, on RoboTwin under the same protocol:
> > >
> > > | |Bottle|Hammer|clock|Bell|Roller|LiftPot|Pillbottle|Container|ObjectStand|PhoneStand|Mean|
> > > |-|-|-|-|-|-|-|-|-|-|-|-|
> > > |RDT|68|40|9|6|43|10|0|20|5|5|20.6(-22.4)|
> > > |Pi0|53|23|11|3|**76**|38|1|45|9|3|26.2(-16.8)|
> > > |**Ours**|**69**|**42**|31|**38**|68|**42**|**46**|**72**|15|**7**|**43.0**|
> > >
> > > Both VLAs achieve lower success rates than Tesseract (33.9%). The results are similar with that reported in the _RoboTwin leaderboard (hard setting)_. This gap arises because VLAs compress perception, spatial understanding, and action generation into a single mapping without explicit future exploration, making them brittle for hard tasks.
> > >
> > > **The results clearly highlight the advantages and potential of the 4D-based planner**.
> > > > **R3. Clarification On Real-Robot and Simulation Performance**
> > >
> > > We fully acknowledge that absolute success rates have room for improvement; this is true for all existing methods. **Our goal is not to present a fully "solved" system for real-world embodied manipulation, as this remains a long-term grand challenge and open question for the field, but rather to introduce a novel 4D paradigm that drives consistent relative improvements over current baselines.** Evaluated in the context of task complexity and baseline comparisons, our relative gains are substantial:
> > >
> > > **Simulation:** On RLBench, we outperform TesserAct by +5.3% and 4DGen by +25.6%. On the hard and challenging RoboTwin benchmark, our 43.0% success rate not only outperforms 4D-based methods but also **nearly doubles the performance of leading VLA models** (Pi0 and RDT). This significant gap demonstrates the clear advantage of the 4D modeling paradigm.
> > >
> > > **Real-World:** Our real-world evaluation is intentionally rigorous (highly randomized object poses, uncurated depth noise) to avoid artificially inflated metrics. We specifically avoided designing a simplified dataset for the sake of inflated metrics; instead, we construct a complex, realistic benchmark to ensure the paradigm's potential is rigorously validated for future work. Under this strict protocol, most baselines (UniPi, 4DGen) fail entirely. Our method achieves consistent relative gains across nearly all tasks against to baselines (Table 4). For example, our S.R. on _Arrange Boxes_ represents a **doubled relative improvement** (7% to 15%).
> > > > **R4. Online vs. Offline Mode and Downstream Control Gains**
> > >
> > > Furthermore, our method is not limited to offline generation. It also supports online mode. Detailed explanation is in **Reviewer ZT91-W3**. In the online mode, we generate the full 4D future, execute a short action chunk, and re-predict the remaining horizon using new observations. This naturally adapts to unexpected dynamic changes.
> > >
> > > The online mode brings an **8–17% improvement** in S.R. across all platforms. However, since the 4D baselines only support offline execution, we reported only offline results for a **fair comparison**. Even if offline mode, ours already demonstrates massive downstream gains, evidenced by Tab. 3 and 4.
> > >
> > > We acknowledge online mode currently faces latency from diffusion sampling speeds. It can be alleviated by faster sampling, or a hybrid reactive architecture in the future, as detailed in our response to **Reviewer ZT91-W3**. As foundational models accelerate, the online mode will become practical.
> > >
> > > _In summary, we believe this direction holds strong promise for future development._

---

### Official Review · Reviewer_X987 · 2026-03-19

**Soundness:** 2
**Presentation:** 3
**Significance:** 2
**Originality:** 2
**Overall Recommendation:** 3
**Confidence:** 3

**Summary:**

I like the problem setup, and I think the direction is interesting. A view-consistent 4D world model for manipulation makes sense, and I appreciate that the paper tries to connect future prediction to actual action inference rather than stopping at generation. The paper is also reasonably clear. That said, I do not think it is quite strong enough yet for acceptance.
My main issue is that the central contribution does not come through clearly. The general pipeline is sensible, but it reads more like a fairly elaborate system than a clean method with one main idea. There is clearly a lot of engineering effort here, but I was left unsure what the core technical step really is.
The strongest part of the paper is the geometry side. On that front, the results are promising and mostly convincing. But the appearance story is much more mixed. On RLBench, the method gets the best FVD, but not the best PSNR or SSIM. On the real-world dataset, geometry improves, but PSNR and SSIM are still not consistently best. So I buy the claim that the method gives better fused geometry. I do not think the paper really supports a broader claim of being uniformly better at generation.
The control story is also not fully clean. The paper wants to argue that better 4D scene prediction leads to better downstream manipulation, but the evidence does not really isolate that. A lot of the gain could be coming from the action inference stack itself, especially the latent optimisation and residual IDM, rather than from the world model. That makes the overarching message feel a bit muddled.

**Compliance With Llm Reviewing Policy:**

Affirmed.

**Key Questions For Authors:**

See review above

**Limitations:**

Yes

**Strengths And Weaknesses:**

See review above

---

> ### Author Rebuttal · Authors · 2026-03-30
>
> > **W1: Unclear Central Contribution.**
>
> We thank the reviewer and clarify: our core contribution is not a collection of independent modules, but a **co-designed framework** that tightly couples 4D world modeling with test-time action inference via trajectory conditioning.
>
> The key insight is using the trajectory latent as a shared medium between generation and control:
>
> **Forward:** the generator uses this latent (alongside cross-view and cross-modality fusion) to synthesize geometrically consistent 4D futures;
>
> **Backward:**, because generation is explicitly conditioned on this latent, we recover actions by back-propagating through the frozen generator to find the optimal trajectory latent matching the predicted future.
>
> Hence, the geometry-aware generator and test-time optimization are two sides of the same coin. They inherently rely on each other to solve the ill-posed nature of inverse dynamics.
>
> The architectural modules (cross-view/cross-modality fusion) are not isolated additions but necessary enablers: without them, multi-view geometries drift, breaking the  3D point-cloud extraction required by the residual IDM.
>
> All in all, our main core technical step is re-architecting a video diffusion backbone so that a trajectory latent can be optimized  through the generated 4D dynamics future to recover physically executable actions. Based on these points, we will thoroughly revise the logics of our manuscript to ensure this central thesis is immediately clear to the reader.
>
> > **W2. Stronger geometry, competitive generation quality**
>
> We thank the reviewer for the careful assessment. We agree that our results do not uniformly dominate all appearance metrics, and will revise the claim to be more precise: MVISTA-4D provides stronger geometry-consistent 4D generation with strong temporal realism and competitive appearance quality.
>
> **Why our PSNR, SSIM is lower** This is somewhat an evaluation artifact. Single-view baselines are evaluated only on their conditioned view, whereas our model additionally synthesizes two unobserved target views, which do not provide any reference, and is evaluated on all views — making pixel-wise fidelity significantly harder. Under a matched conditioned-view-only protocol,  our appearance quality becomes competitive with the strongest baselines:
>
> |Setting|RLBench PSNR|RLBench SSIM|RoboTwin PSNR|RoboTwin SSIM|Real-world PSNR|Real-world SSIM|
> |-|-|-|-|-|-|-|
> |Best prior baseline|**23.88**|**92.8**|22.98|90.2|**22.53**|91.50|
> |Ours (all generated views)|23.31|90.8|22.91|90.2|21.82|89.98|
> |Ours (conditioned view only)|23.86|92.0|**23.12**|**90.8**|22.12|**91.58**|
>
> > **W3** **Unclear attribution of downstream manipulation gains**
>
> We thank the reviewer for raising this important point. We agree that Table 3 alone does not fully isolate the contribution of the 4D world model from that of the action inference stack, and we will revise the paper to make this distinction clearer. In particular, Table 3 mainly shows the contribution of the action inference components. Our intended claim is therefore not that all downstream control gains come solely from the action inference design, but that improved 4D scene prediction and action inference contribute complementarily to that.
>
> **First**, we do believe the gains are not only from latent optimization or the residual IDM, and that the 4D world model itself also plays an important role. First, in an additional controlled experiment conducted during the rebuttal period, we replace our RGBD generation with RGB-only generation while keeping the same action inference pipeline unchanged. Under this setting, performance drops from **72.6 to 69.5** on RLBench and from **43.0 to 38.7** on RoboTwin. Since the action inference stack is fixed, this result suggests that geometry-aware 4D RGBD generation itself contributes to downstream manipulation.
>
> **Second**, our existing multi-view ablation also supports this interpretation. In **Appendix Table 9**, with the same downstream action inference mechanism, increasing the number of generated views improves RLBench success from 68.6 (1 view) to 72.6 (3 views). This indicates that stronger multi-view 4D prediction provides more complete geometric information and more reliable cues for control. This is also consistent with the improved depth and fused point-cloud metrics in **Tables 1 and 2**.
>
> We will therefore clarify the paper’s message as follows:** the downstream improvement is driven by the interaction between **a stronger 4D world model and a stronger action inference module**, and the world model contributes by providing a more complete and geometrically consistent predictive substrate for manipulation.

---

### Decision · Program_Chairs · 2026-04-30

**Decision:**

Accept (regular)

**Comment:**

This paper proposes  a multi-view 4D world model for robotic manipulation that generates view-consistent RGB-D via test-time latent optimization. It received two Weak Reject, one Weak Accept, and one Accept . Positive reviewers appreciate the interesting “imagine-then-act” paradigm and the good geometric consistency results. The negative reviewers raised concerns about system complexity, low absolute real-world success rates, and inference latency. The rebuttal addressed most of the concerns with new ablations, and the authors clarified that the method’s goal is to establish a new 4D paradigm rather than a production‑ready system. The AC acknowledges that the paper’s real-world embodied impact is not yet fully established, but considers it a valuable step forward in video generation and 4D world modeling. Given the technical novelty, and the field’s need for such exploration, the AC recommends Accept.